# DIAGNOSING AND MITIGATING SYSTEM BIAS IN SELF-REWARDING RL

## ABSTRACT

Reinforcement learning with verifiable rewards (RLVR) efficiently scales the reasoning ability of large language models (LLMs) but remains bottlenecked by limited labeled samples for continued data scaling. Reinforcement learning with intrinsic rewards (RLIR), in which the policy model assigns reward signals to its own rollouts, enables sustainable scaling in unlabeled settings. Yet its performance and stability still lag behind RLVR. We trace this gap to a system bias: the model tends to deem its own high-confidence rollouts correct, leading to biased and unstable reward estimation. It accumulates and rises rapidly as training proceeds, with the deviation from the oracle drifting toward over-reward. This causes unstable training and locks the performance ceiling. To understand how system bias yields these effects, we characterize it by the magnitude of reward bias, the degree of policy–reward coupling, and the proportional imbalance between over-reward and under-reward via three metrics: $\rho_{\text{noise}}$, $\rho_{\text{selfbias}}$, and $\rho_{\text{symbias}}$. We find that $\rho_{\text{noise}}$ and $\rho_{\text{symbias}}$ affect convergence performance and speed, while $\rho_{\text{selfbias}}$ has an amplification effect: it amplifies both correct and incorrect updates and induces unstable reward estimation. To mitigate system bias of RLIR, we propose reinforcement learning with ensembled rewards (RLER). It aggregates diverse models with adaptive reward interpolation and rollout selection strategy to build a unified reward-estimation space, jointly improving accuracy ($\rho_{\text{noise}}$), unbiasedness ($\rho_{\text{selfbias}}$, $\rho_{\text{symbias}}$), and robustness ($\rho_{\text{selfbias}}$). Extensive experiments show that RLER improves by **+13.6%** over the best RLIR baseline, and is only **3.6%** below the RLVR setting. Moreover, RLER achieves stable scaling on unlabeled samples, making it highly applicable.

## 1 INTRODUCTION

Reinforcement learning with verifiable rewards (RLVR) can efficiently scale the reasoning capabilities of large language models (LLMs) (Guo et al., 2025; El-Kishky et al., 2025; Team et al., 2025; Gao et al., 2023). However, it is bottlenecked by the scarcity of labeled data, limiting continued data scaling (Gunjal et al., 2025; Zhang et al., 2025c). In contrast, reinforcement learning with intrinsic rewards (RLIR, also known as self-rewarding RL), in which the policy model assigns reward signals to itself, enables sustainable scaling in unlabeled settings (Huang et al., 2025; Zuo et al., 2025), reducing annotation cost and potentially enabling models to reach higher capability levels. It is also well suited to scenarios with scarce annotation, private corpora, or industrial settings that have abundant unlabeled data and require rapid iteration.

Nevertheless, its performance gain and stability still fall short of RLVR (Shafayat et al., 2025; Zhang et al., 2025c). Our analysis shows that under RLIR, the model tends to deem its own high-confidence rollouts correct. This induces system bias, manifested as biased and unstable reward estimation. Specifically, the magnitude of this estimation bias is highly correlated with rollout correctness and confidence: it is small for confident correct rollouts but large for confident mistakes. Under existing RLIR methods (Zuo et al., 2025; Huang et al., 2025), we find that the reward-estimation bias accumulates and rises rapidly as training proceeds, with the deviation from the oracle drifting toward over-reward, leading to unstable training and tightly locking the performance ceiling. To understand how system bias yields these effects, we characterize it via three metrics: (i) **reward noise rate** $\rho_{\text{noise}}$: the magnitude of reward bias. (ii) **self-feedback bias rate** $\rho_{\text{selfbias}}$: the coupling strength be-

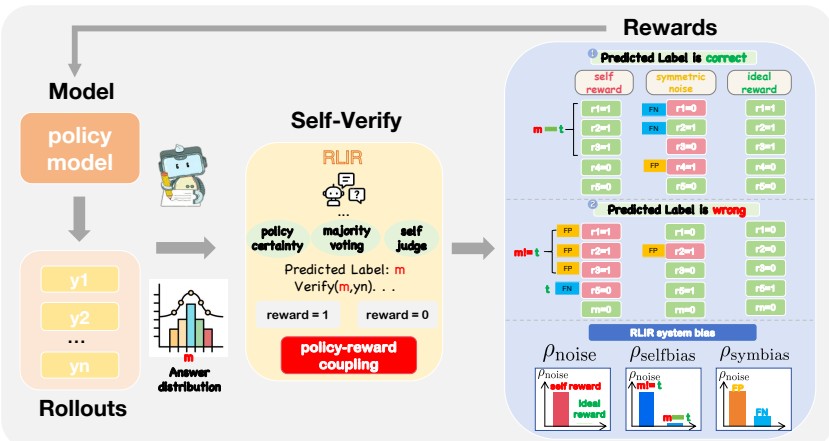

Figure 1: Flowchart of the process for reinforcement learning with intrinsic rewards (RLIR).

tween the policy answer distribution and the reward distribution. (iii) **symmetry bias rate** $\rho_{\text{symbias}}$: the proportional imbalance between over-reward and under-reward.

Based on the three metrics above, we conduct bottom-up analytical experiments and obtain the following insights: (i) $\rho_{\text{noise}}$ governs both the convergence performance and the convergence rate; when excessive, it can even cause training collapse. (ii) The $\rho_{\text{symbias}}$ metric indicates that over-reward is more detrimental than under-reward. (iii) High $\rho_{\text{selfbias}}$ amplifies both correct and incorrect updates. High-confidence rollouts receive higher rewards: when correct, it strengthens alignment; when incorrect, it amplifies wrong-direction updates. (iv) High $\rho_{\text{selfbias}}$ induces unstable reward estimation: prediction correctness exhibits large across-instance variance, this variance propagates through policy-reward coupling, yielding unstable reward estimation.

Therefore, to achieve stable unlabeled data scaling, the reward-estimation space should simultaneously satisfy: (i) **Accuracy**: low $\rho_{\text{noise}}$ kept below the collapse threshold. (ii) **Unbiasedness**: reduced over-reward ($\rho_{\text{symbias}}$) and weak policy–reward coupling on incorrect rollouts ($\rho_{\text{selfbias}}^{\text{err}}$). (iii) **Robustness**: stable reward estimates under policy–reward coupling ($\rho_{\text{selfbias}}$).

To mitigate the system bias in single-policy models of RLIR, we propose reinforcement learning with ensembled rewards **(RLER)**. RLER adopts a population-based strategy: it replaces single–model self-rewarding with an ensemble, aggregating diverse models to construct a unified stable reward space that guides the ensemble to improve collaboratively. We optimize the sub-objectives via: (i) **Ensemble Self-Rewarding**: jointly achieving accuracy, unbiasedness, and robustness. (ii) **Adaptive Soft-reward Interpolation**: adjusting the weight between hard and soft rewards according to unified confidence, balancing accuracy and robustness. (iii) **Confidence–disagreement Balanced Rollout Selection**: down-weighting high-confidence errors while retaining scarce correct samples, improving accuracy and unbiasedness. Finally, we apply model merging to consolidate the ensemble into a single deployable model for practical use.

To systematically evaluate RLER, we conduct extensive experiments. The results show that RLER improves by +13.6% over the best RLIR baseline, and is only 3.6% below the RLVR setting. More importantly, RLER effectively mitigates the impact of system bias, greatly optimize $\rho_{\text{noise}}$, $\rho_{\text{selfbias}}$, and $\rho_{\text{symbias}}$. Finally, we observe stable scaling with unlabeled data, via model merging, the deployable model has higher accuracy and stability.

## 2 RELATED WORKS

**Reinforcement learning with intrinsic rewards (RLIR)** RLIR dispenses with human labels by having the model generate outputs as policy rollouts, and provide rewards through a rollout-based reward estimation rule or self-judging mechanism. Methods cluster into three families: (i) Self-consistency: majority-vote across policy answers to obtain a predicted label, then verify to obtain rewards (Zuo et al., 2025; Huang et al., 2025); (ii) Probability–based: which use the policy's entropy (Zhang et al., 2025a; Agarwal et al., 2025) or certainty (Li et al., 2025; Zhao et al., 2025) to assign rewards directly; and (iii) LLM-as-a-judge: through self-judge/play to improve verifiability and

coverage (Arnesen et al., 2024; Yuan et al., 2024; Xiong et al., 2025). The first two families are internally aligned on their objective: maximizing answer-distribution agreement, while differing in how sharply they refine the answer-distribution to reward distribution (Li et al., 2025; Zhang et al., 2025b). Yet they typically rely on a single policy model, which tightly couples the reward to the current policy and locks the performance ceiling. They also lack adaptive reward estimation rules, for example a unified treatment of "hard vs. soft rewards" and "retain vs. prune rollouts," which results in instability.

**Learning with Noisy Labels** Learning with noisy labels aimed at improving model robustness under noise (Frénay & Verleysen, 2013; Zhang et al., 2016a; Nigam et al., 2020). Based on the dependence on features, label noise is typically divided into instance-independent noise and instance-dependent noise. The former further includes symmetric (equal flip probability across classes) and asymmetric noise (Song et al., 2022; Zhang et al., 2016b). In contrast, the reward noise in RLIR is not a simple symmetric or instance-dependent label noise. It stems from the policy model's system bias and manifests as a strong coupling between the reward distribution and the model's predictive distribution, together with a over-reward/under-reward noise imbalance.

## 3 PRELIMINARY

In this section, we start by introducing the working process of RLIR. Subsequently, we characterize the system bias from three aspects: reward bias magnitude, policy-reward coupling strength, imbalance magnitude between over-reward and under-reward to assess its impact on training.

### 3.1 PROCESS OF RLIR

In general, RLIR consists of three stages: *step 1.* a query $x \in \mathcal{X}$ is fed to policy model $\pi_\theta(y_{1:T} \mid x) = \prod_{t=1}^{T} \pi_\theta(y_t \mid x, y_{<t})$ to sample rollouts $\mathcal{Y}_\theta(x)$; *step 2.* self-rewards are estimated from the rollouts: $\mathcal{R}(\mathcal{Y}_\theta(x))$; *step 3.* the rewards are converted to advantages $A$, which are then used to compute policy gradients $\nabla_\theta \mathcal{L}(\theta)$ and update the policy.

We instantiate GRPO (Shao et al., 2024); the group-based self-reward estimator is:

$$\{r_i\}_{i=1}^{G} = \mathcal{R}\big(\mathcal{Y}_\theta(x)\big).$$

where $G$ denotes the group size, $r_i$ denotes the estimated reward of the $i$-th rollout $y_i$. As concrete baselines, we consider Self-Consistency (SC) and Frequency-based (Freq). Define the labeling map $\ell : \mathcal{Y} \to \{0, \ldots, L-1\}$. SC estimates the answer distribution via empirical frequencies $p_j = \frac{1}{G} \sum_{i=1}^{G} \mathbf{1}[\ell(y_i) = j]$ and take predicted label $m = \arg\max_j p_j$, while Freq assigns each rollout the corresponding empirical class probability.

$$\mathcal{R}_{\mathrm{SC}}\big(\mathcal{Y}_\theta(x)\big) = \big\{\mathbf{1}[\ell(y_i) = m]\big\}_{i=1}^{G}, \qquad \mathcal{R}_{\mathrm{Freq}}\big(\mathcal{Y}_\theta(x)\big) = \big\{p_{\ell(y_i)}\big\}_{i=1}^{G}.$$

The RL objective is to maximize the expected group reward and parameters $\theta$ are updated via gradient ascent:

$$\max_\theta \mathcal{J}(\theta) = \mathbb{E}_{x \sim \mathcal{X}, \, \mathcal{Y}_\theta(x) \sim \pi_\theta} \left[\frac{1}{G} \sum_{i=1}^{G} r_i\right], \quad \theta \leftarrow \theta + \eta \nabla_\theta \mathcal{J}(\theta).$$

### 3.2 REWARD NOISE RATE

Let $t$ be the ground-truth label. For each rollout $y_i$, define the oracle reward $r_i^\star = \mathrm{verify}(\ell(y_i), t)$ and the attained reward as $r_i$. To quantify the magnitude of reward bias between $r_i$ and $r_i^\star$, we define the *noise rate* as:

$$\rho_{\mathrm{noise}}(x) = \frac{1}{G} \sum_{i=1}^{G} \big| r_i - r_i^\star \big|.$$

### 3.3 SELF-FEEDBACK BIAS RATE

RLIR induces *policy–reward coupling*: the policy's answer distribution shapes the reward distribution. With self-estimated reward $\tilde{r}_i$, we quantify this coupling by the *self-feedback bias rate*:

$$\rho_{\mathrm{selfbias}}(x) = 1 - \frac{1}{G} \sum_{i=1}^{G} |r_i - \tilde{r}_i|, \qquad \rho_{\mathrm{selfbias}}^{\mathrm{SC}}(x) = 1 - \frac{1}{G} \sum_{i=1}^{G} \Big| r_i - \mathbf{1}\big(\ell(y_i) = m(x)\big) \Big|.$$

**Correctness–confidence effect** $\rho_{\text{noise}}$ is highly correlated with rollout correctness and confidence under RLIR. Let $p_t$ and $p_m$. *(i)* $m = t$. Under hard-reward: $\rho_{\text{noise}} = 0$. For soft-reward, the deviation from the oracle shrinks with confidence: $\left| \mathbb{E}[r] - \mathbb{E}[r^\star] \right| \leq 1 - p_m$. *(ii)* $m \neq t$. $\rho_{\text{noise}} = p_t + p_m$ and the misupdate strength grows with the margin $(p_m - p_t)$, hence higher confidence $p_m$ worsens the reward bias. Soft-reward weakens it by distributing credit, we prove that the attenuation is stronger when the non-majority distribution is dispersed:

**Theorem 1.** *If $p_t \geq \max_{j \notin \{m,t\}} p_j$, soft-rewards are closer to the oracle than hard-rewards (details seen in Appendix A).*

### 3.4 SYMMETRY-BIAS RATE

Compared to symmetric noise, RLIR's policy–reward coupling introduces a proportional imbalance between over-reward and under-reward. We term the directional components false-negative (FN): under-reward relative to the oracle; and false-positive (FP): over-reward relative to the oracle. With $(u)_+ = \max\{u, 0\}$,

$$\text{FN}(x) = \frac{1}{G} \sum_{i=1}^{G} \left( r_i^\star - r_i \right)_+, \qquad \text{FP}(x) = \frac{1}{G} \sum_{i=1}^{G} \left( r_i - r_i^\star \right)_+,$$

With oracle accuracy $\Pr(r^\star = 1)$, balance ratio under RLIR and under symmetric noise:

$$\text{BR}_{\text{IR}}(x) = \frac{\text{FN}(x)}{\text{FP}(x)}, \qquad \text{BR}_{\text{sym}} = \frac{\Pr(r^\star = 1)}{1 - \Pr(r^\star = 1)}.$$

We define the *symmetry bias rate* as the deviation of the $\text{BR}_{\text{IR}}(x)$ from $\text{BR}_{\text{sym}}$:

$$\rho_{\text{symbias}}(x) = \text{BR}_{\text{IR}}(x) - \text{BR}_{\text{sym}}.$$

### 3.5 DECOUPLING EXPERIMENT

We conduct a systematic set of experiments to separately analyze the effects of three metrics on RLIR training and to identify the causes of biased and unstable reward estimation.

**Experiment setting** To achieve strong control over the experiment and ensure there is no data contamination, we synthesize an arithmetic dataset (375k) with operators $\{+, -, //, \%\}$ (See Appendix B.1 for details). QWEN2.5-1.5B-INSTRUCT is used as base policy $\pi_\theta$.

We isolate the three metrics $\{\rho_{\text{noise}}, \rho_{\text{symbias}}, \rho_{\text{selfbias}}\}$ via a controlled construction from oracle rewards $\{r_i^\star\}$ by first injecting symmetric noise to control the noise rate ($\rho_{\text{noise}}$), then applying asymmetric flipping to adjust the FN/FP balance ($\rho_{\text{symbias}}$), and finally coupling the rewards with model predictions to modulate self-feedback strength ($\rho_{\text{selfbias}}$). Then, we test different RLIR methods in empirical training and observe the following insights:

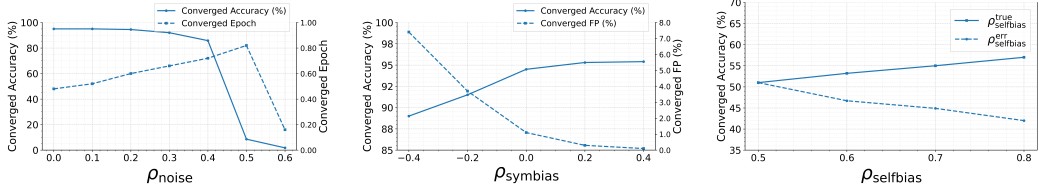

Figure 2: Effects of $\rho_{\text{noise}}$, $\rho_{\text{symbias}}$, and $\rho_{\text{selfbias}}$ during training on the arithmetic dataset.

**Findings 1: $\rho_{\text{noise}}$ governs the convergence performance and speed.** As $\rho_{\text{noise}}$ rises, the performance ceiling drops and training shifts from stable convergence to collapse; within the transition regime, higher noise monotonically slows convergence.

**Findings 2: Over-reward is more detrimental than under-reward.** With $\rho_{\text{noise}}$ held constant, as $\rho_{\text{symbias}}$ increases, the imbalance shifts from an over-reward bias to an under-reward bias; meanwhile, the converged performance rises, indicating that over-rewarding is more detrimental. Further analysis shows that under-reward weakens the gradient along the correct direction, whereas over-reward assigns positive advantage to incorrect outputs; both effects dampen correct updates and introduce a near-orthogonal gradient bias (as seen in Fig. 2(b) and Fig. 3(e)).

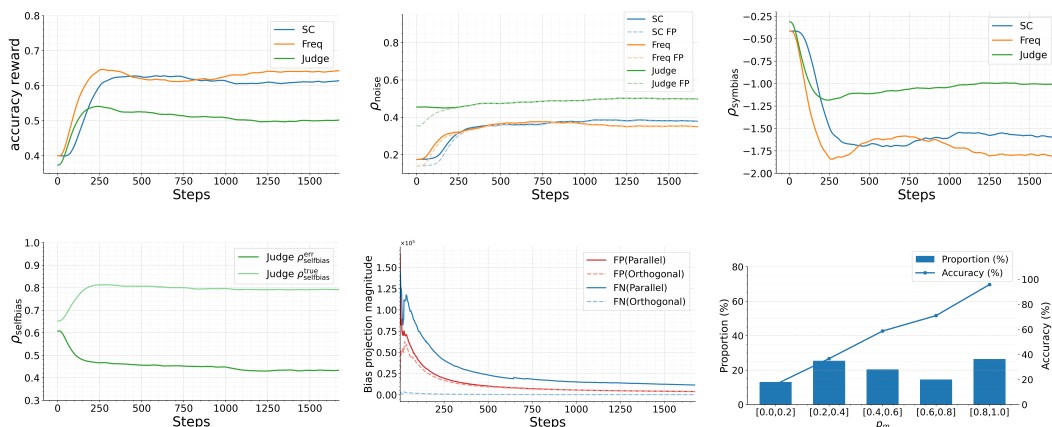

Figure 3: Training results of RLIR methods on arithmetic dataset.

**Findings 3: High $\rho_{\text{selfbias}}$ amplifies both correct and incorrect updates.** As seen in Figure 2(c), when $m = t$, a higher $\rho_{\text{selfbias}}^{\text{true}}$ strengthens correct updates, leading to improved convergence performance; when $m \neq t$, $\rho_{\text{selfbias}}^{\text{err}}$ amplifies wrong-direction updates.

As seen in Figure 3, under RLIR methods, we find that the reward-estimation bias accumulates and rises rapidly as training proceeds ($\rho_{\text{noise}}$, seen in Fig. 3(b)), with the deviation from the oracle drifting toward over-reward ($\rho_{\text{symbias}}$, seen in Fig. 3(c)), tightly locking the performance ceiling (seen in Fig. 3(a)). As for policy–reward coupling, SC and Frequency yield $\rho_{\text{selfbias}} \equiv 1$. Judge achieves $\rho_{\text{selfbias}} < 1$; however, $\rho_{\text{selfbias}}^{\text{true}}$ is low to strengthen correct updates, whereas $\rho_{\text{selfbias}}^{\text{err}}$ remains high enough to lock in updates in the wrong direction (seen in Fig. 3(d)).

**Findings 4: High $\rho_{\text{selfbias}}$ induces unstable reward estimation.** Prediction correctness and confidence ($p_m$) exhibits large cross-instance variance (seen in Fig. 3(f)). We just found that RLIR methods exhibit very high policy–reward coupling, the variance propagates through this coupling, yielding unstable reward estimation.

**What reward space do we need?** Therefore, to achieve stable unlabeled scaling, the reward-estimation space should simultaneously satisfy: (i) *Accuracy:* a low noise rate, with $\rho_{\text{noise}}$ kept below the collapse threshold; (ii) *Unbiasedness:* reduced over-reward bias ($\rho_{\text{symbias}}$) and weak policy–reward coupling on incorrect rollouts ($\rho_{\text{selfbias}}^{\text{err}}$); (iii) *Robustness:* it keeps the reward estimate stable, under policy–reward coupling ($\rho_{\text{selfbias}}$).

## 4 RLER

Building on the diagnostics above, to mitigate the system bias in single-policy models of RLIR, we propose reinforcement learning with ensembled rewards (**RLER**), which jointly improves **accuracy**, **unbiasedness**, and **robustness**.

### 4.1 ENSEMBLE SELF-REWARDING

We replace single-model self-rewarding with an ensemble, aggregating diverse models to construct a unified reward space that guides the ensemble to improve collaboratively.

**Aggregation.** Given $K$ source policy models $\{\pi_{\theta_k}\}_{k=1}^{K}$, draw rollouts $\mathcal{Y}_k(x) \in \mathcal{S}\big(\pi_{\theta_k}(\cdot \mid x)\big)$ for each $k$, and denote the answer of a rollout by $\ell(y)$. Let the per–source answer distributions be $p_j^{(k)} := \Pr_{\pi_{\theta_k}}\big[\ell(y) = j \mid x\big]$. Define the ensemble mixture:

$$\bar{p}_j := \frac{1}{K} \sum_{k=1}^{K} p_j^{(k)}, \qquad m^{\text{EC}} := \arg\max_j \bar{p}_j, \qquad \mathcal{Y}(x) := \bigcup_{k=1}^{K} \mathcal{Y}_k(x)$$

**Why ensemble first.** By convexity of $\max$, $\bar{p}_t - \max_{j \neq t} \bar{p}_j \geq \frac{1}{K} \sum_k (p_t^{(k)} - \max_{j \neq t} p_j^{(k)})$; averaging thus nudges negative/fragile margins toward zero, reducing single-source mistakes and lowering $\mathbb{E}[r_{\text{noise}}]$. Using the mixture $\bar{p}$ also weakens single-policy coupling ($\rho_{\text{selfbias}}$) and disperses

error mass across classes (lower over-reward skew, i.e., $\rho_{\text{symbias}}$), while aggregating rewards across sources smooths estimates against confidence swings, improving robustness.

## 4.2 ADAPTIVE SOFT-REWARD INTERPOLATION

To mitigate misestimation caused by hard rewards and the low-confidence bias inherent in soft rewards, we propose an adaptive interpolation strategy that dynamically adjusts the hard/soft weighting, seeking ***the optimal trade-off between accuracy and robustness***.

**Interpolation.** Let the ensemble hard and soft rewards be $r_i^{\text{H}} = \mathbf{1}\big[\ell(y_i) = m^{\text{EC}}\big]$ and $r_i^{\text{S}} = \bar{p}_{\ell(y_i)}$. We interpolate by:

$$r_i^{(\alpha)} = (1 - \alpha)\, r_i^{\text{H}} + \alpha\, r_i^{\text{S}}, \qquad \alpha \in [0, 1].$$

**Unified Answer-Confidence Distribution Estimation** For each source $k \in \{1, \dots, K\}$, let $\mathcal{Y}_k(x)$ denote its rollouts, and let $\mathcal{Y}_{k,j}(x) \subseteq \mathcal{Y}_k(x)$ denote those with answer $j = \ell(y)$. Define $\ell_{k,y}$ as the average token probability of rollout $y$ from source $k$. Denote the per-source answer frequency and confidence by:

$$P_k(j) = \frac{|\mathcal{Y}_{k,j}(x)|}{|\mathcal{Y}_k(x)|}, \qquad \bar{\ell}_k(j) = \frac{1}{|\mathcal{Y}_{k,j}(x)|} \sum_{y \in \mathcal{Y}_{k,j}(x)} \ell_{k,y}\,.$$

Let $\big(L_{\min}^{(k)}, L_{\max}^{(k)}\big)$ be the batch-wise answer-confidence bounds for source $k$ at the current step. We linearly normalize the answer confidence by these bounds:

$$C_k(j) = \frac{\bar{\ell}_k(j) - L_{\min}^{(k)}}{L_{\max}^{(k)} - L_{\min}^{(k)}}\,.$$

This injects information about the sample's relative difficulty within the batch, enhancing the accuracy and robustness of the estimate. Combine frequency and confidence within each source, we then renormalize within the group to align the scales:

$$S_k(j) = P_k(j)\, C_k(j), \qquad s_k(j) = \frac{S_k(j)}{\sum_u S_k(u)}\,.$$

Finally, we aggregate across sources to obtain a accurate and robust answer-confidence unified estimation and the predicted-label confidence:

$$\tilde{p}_j(x) = \frac{1}{K} \sum_{k=1}^{K} s_k(j), \qquad \alpha(x) = \text{clip}\Big(\tilde{p}_{m^{\text{EC}}}(x),\, 0,\, 1\Big).$$

## 4.3 CONFIDENCE–DISAGREEMENT BALANCED ROLLOUT SELECTION

To further ***improve accuracy and unbiasedness***, we select updates from the pooled rollouts, suppress gradient contamination caused by single-source reward bias.

**Rollout allocation strategy** We treat all ensemble rollouts as one data pool and allocate updates to the $K$ sources in two ways:

- **Data sharding.** Partition the query set as $\mathcal{Q} = \bigcup_{k=1}^{K} \mathcal{Q}_k$. Model $k$ updates on queries $x \in \mathcal{Q}_k$ using the pooled rollouts $\mathcal{Y}(x) = \bigcup_{j=1}^{K} \mathcal{Y}_j(x)$.

- **Model sharding.** For each query $x$, split the pooled rollouts $\mathcal{Y}(x)$ evenly across models for updates.

Experiments show that data sharding provides stronger diversity, we therefore use it by default.

**Rollout selection strategy** Partition answer distribution into the head $m^{\text{EC}}$ and the tail $\mathcal{L} \backslash \{m^{\text{EC}}\}$.

$$w_{m^{\text{EC}}}(x) = \alpha(x),$$
$$w_j(x) = 1 - \tilde{p}_j(x), \qquad j \neq m^{\text{EC}}.$$

Let $n_y$ assign per-answer quotas and the dynamic per-question budget is:

$$\text{take}_y = \min\Big\{ n_y,\, \text{round}\big(n_y \cdot w_y(x)\big) \Big\}, \qquad b(x) = \sum_y \text{take}_y$$

This allocation makes the head budget contract/expand with the confidence gate $\alpha(x)$. Meanwhile, tail both effectively suppress low-confidence tail reward bias and, when $m^{\text{EC}} \neq t$, concentrate sampling on the minority true label, amplifying its corrective signal with reward interpolation.

## 4.4 ENSEMBLE-TO-SINGLE CONSOLIDATION

To enhance practical applicability, we finally apply model merging (Ties-Merging (Yadav et al., 2023), $k = 0.7$, $\alpha = 0.5$) to consolidate the ensemble into a single model, resolving the multi-model deployment issue.

## 5 EXPERIMENTS

Centered on RLER, §5.2 empirically compares it to baselines and validates the three desiderata—accuracy, unbiasedness, and robustness; §5.3 explores its best-performing variants; §5.4 demonstrates its practical value.

### 5.1 EXPERIMENTAL SETTINGS

**Models.** We conduct experiments on the Qwen2.5 Series (Yang et al., 2024b;a), using variants at different scales. Unless otherwise noted, the default model is QWEN2.5-MATH-7B.

**Datasets and Benchmarks.** We train on two corpora: our arithmetic dataset and DAPO-MATH-17K (Yu et al., 2025). For the arithmetic dataset, we evaluate on a 500-problem in-distribution test set. For DAPO-MATH-17K, we train QWEN2.5-MATH-7B and evaluate on six challenging benchmarks: MATH500 (Hendrycks et al., 2021), AMC23 (Li et al., 2024), AMC24, AIME24 (Li et al., 2024), AIME25 (MAA, 2024), and HMMT24. We report both Avg@k and Pass@k to ensure robust and comprehensive evaluation.

**Baselines.** We compare against methods covering both hard-reward and soft-reward paradigms: hard-reward Self-Consistency and LLM-as-a-Judge, the soft-reward Frequency-Based approach.

**Details.** We use the Open-R1 framework and apply GRPO. For DAPO-MATH-17K, we set the number of rollouts to $G$=16 and use an ensemble of $k = 2$ sub-policy models; consequently, per-policy rollouts are $G_k$=8 for fair comparison. See Table 2 for results with larger $k$. Other hyperparameters are as follows: the learning rate $1\times10^{-6}$, the KL regularization coefficient $\beta$=0.001, the sampling temperature 0.9. Details and results on the arithmetic dataset are provided in Appendix B. Prompt templates are provided in Appendix C.

### 5.2 MAIN RESULTS

**Accuracy.** The results of compared methods on DAPO-MATH-17K and benchmarks are shown in Figure 4 and Table 1. In terms of performance, JUDGE $<<$ FREQ $<$ SC $<$ RLER $\lesssim$ RLVR. RLER attains 96.0% test accuracy relative to RLVR, representing an average improvement of +45.9% over pretraining and +13.6% over the best RLSR baseline. To explain the performance differences, we quantitatively analyze the metrics we define in § 3. As shown in Figure 4, RLER significantly reduces $\rho_{noise}$ during training, accuracy rise steadily and closely track RLVR.

**Unbiasedness.** To further understand RLER's performance improvement, we find that RLER markedly suppresses the highly harmful FP component of $\rho_{noise}$ and effectively alleviates the negative skew in $\rho_{symbias}$. This prevents the model from falling early into the "over-reward bias amplification" trap and raises the attainable performance ceiling. The improvement stems from (i) rollout

Table 1: Zero-shot Avg@8 and Pass@8 of QWEN2.5-MATH-7B across six reasoning benchmarks.

| Method | AIME24 Avg@8 | AIME25 Avg@8 | AMC23 Avg@8 | AMC24 Avg@8 | MATH Avg@8 | HMMT24 Avg@8 | Avg. Avg@8 | Avg. Pass@8 |
|---|---|---|---|---|---|---|---|---|
| w/o RL | 12.5 | 6.40 | 45.3 | 23.0 | 59.2 | 7.9 | 25.7 | 54.8 |
| *RLVR* | 32.1 | 12.5 | 65.0 | 34.2 | 79.1 | 10.4 | 38.9 | 55.5 |
| *RLIR* LLM-as-a-Judge | 3.3 | 1.7 | 23.1 | 18.4 | 34.1 | 0.0 | 13.4 | 22.5 |
| Self-Consistency | 16.3 | **13.8** | 55.9 | 32.8 | 75.0 | 4.2 | 33.0 | 47.1 |
| Frequency-Based | 11.7 | 8.8 | 43.1 | 25.8 | 71.7 | 1.7 | 27.1 | 31.6 |
| *RLER* | **23.3** | 12.1 | **66.9** | **35.8** | **77.5** | **9.6** | **37.5** | **52.8** |

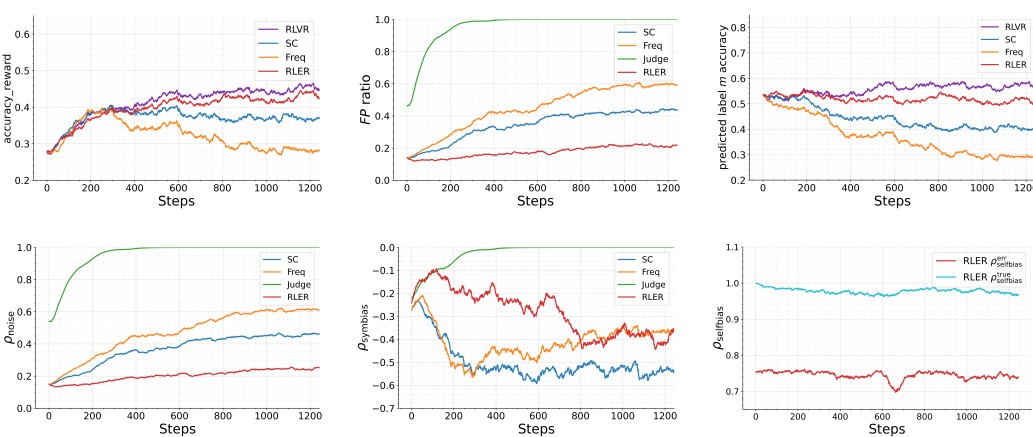

Figure 4: Training results of compared baselines and RLER on DAPO-MATH-17K.

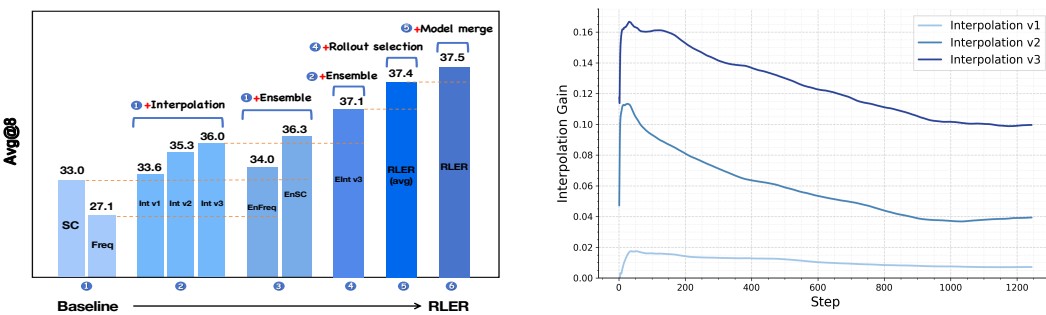

Figure 5: Ablations of RLER (left) and reward interpolation (right). Left: Avg@8 on the test benchmarks for each ablation. Right: interpolation gain across interpolation variants.

allocation, which increases ensemble diversity, and (ii) rollout selection, which removes over-reward bias and, in conjunction with reward interpolation, rectifies under-reward bias, all (iii) within the ensemble unified reward space. As a result, RLER no longer relies on a single model's system bias, mitigating biased reward estimation. Empirically, $\rho_{\text{selfbias}}^{\text{true}} \approx 1$ while $\rho_{\text{selfbias}}^{\text{err}}$ drops substantially.

**Robustness.** As noted in Section 3.5, strong policy–reward coupling with large variance of prediction correctness and confidence broadly amplifies reward bias. RLER, via *Reward Interpolation*, naturally filters low-confidence bias, while the ensemble's unified reward space counteracts the single model's high-confidence bias. Maj@k (the accuracy of predicted label $m$) and Pass@k respectively reflect the "correctness of the most confident answer" and the "existence of a correct answer." Results show that, for compared methods, Pass@k drops markedly before and after training, Maj@k decreases monotonically during training, which reflects contamination from erroneous updates, whereas RLER demonstrates strong robustness.

### 5.3 VARIANTS ABLATIONS

**Ensemble Self-Rewarding.** We assess the contribution of each component by ablating *Model Merge*, *Rollout Selection*, *Reward Interpolation*, and *Ensemble* from RLER individually. As shown in Figure 5, the pronounced degradation when removing the *Ensemble* indicates that mitigating system bias to improve accuracy is the most critical factor. Furthermore, we find that performing *Reward Interpolation* within the ensemble space yields superior performance. We hypothesize that this stems from the ensemble's unified reward space: diversity across models reduces $\rho_{\text{selfbias}}^{\text{err}}$, improves the robustness of the reward space, and enables $\alpha(x)$ to be estimated more accurately and stably within the ensemble space.

**Adaptive soft-reward interpolation.** As shown in Figure 5, removing *Reward Interpolation* leads to a substantial performance drop. We further analyze the necessity of each component in our interpolation method: starting from *Int v3* (ours), dropping the batch-wise linear normalization $C_k$ and the group-wise confidence distribution renormalization $s_k$ yields *Int v2*; further removing the

| Select method | $m = t$ | $m \neq t$ | |
|---|---|---|---|
| | $b_{\text{avg}}$ | $\rho_{\text{noise}}$ | $b_{\text{avg}}$ |
| select_all | 16.0 | 65.5 | 16.0 |
| m_only | 12.1 | 100.0 | 9.9 |
| m_except | 3.9 | 8.7 | 6.1 |
| ours | **12.0** | **50.5** | **11.3** |

Figure 6: Rollout Allocation (left) and Rollout Selection (right) across compared methods and RLER. Left: diversity gain across allocation methods. Right: average selected rollouts ($b_{\text{avg}}$) and reward noise ($\rho_{\text{noise}}$) conditioned on $m^{\text{EC}}$ correctness.

confidence estimate $\bar{\ell}_k$ produces *Int v1*, where we instead control the interpolation strength via annealing (with $\alpha$ decaying over training steps). We measure the interpolation gain as $|r^{\text{H}} - r^{\star}| - |r^{(\alpha)} - r^{\star}|$. The results show that Int v3 attains the best performance and the largest interpolation gain, confirming the contribution of each step.

**Confidence–disagreement balanced rollout selection.** We demonstrate the advantages of our approach under the *Rollout Allocation* and *Rollout Selection* strategies, shown in Figure 6. For the allocation strategy, we quantify the diversity gain by the accuracy gap between the ensemble $m^{\text{EC}}$ and the average individual model: $\Delta_{\text{div}} = \text{Acc}(m^{\text{EC}}) - \frac{1}{M}\sum_{i=1}^{M} \text{Acc}(m_i)$. Results show that *Data Sharding* yields a larger $\Delta_{\text{div}}$. For the selection strategy, we measure the average number of selected rollouts $b_{\text{avg}}$ and the reward noise rate $\rho_{\text{noise}}$ conditioned on whether $m^{\text{EC}}$ is correct. Here, *m only* selects only $m^{\text{EC}}$, while *m except* excludes $m^{\text{EC}}$. Our method exhibits a higher selection rate when $m^{\text{EC}} = t$ and effectively discards FP samples when $m^{\text{EC}} \neq t$, reducing $\rho_{\text{noise}}$ compared to *select all*. These results validate that our *Rollout Selection* improves both accuracy and unbiasedness.

## 5.4 PRACTICAL VALUE OF RLER

**Stably Scalable Unlabeled RL.** In real-world scenarios, the absence of human-annotated labels and limited resources mean we cannot know a priori: how much data is needed to reach optimal performance, thus, a stably scalable unlabeled RL algorithm is crucial. To assess the practical value of RLER, we examine its performance across different data sizes (adding Big-Math (Albalak et al., 2025) for further scaling), as shown in Figure 7. Compared with RLIR methods, RLER exhibits stably scalable behavior from 8k to 1024k. Notably, the merged model not only resolves the multi-model deployment issue but also achieves higher accuracy and stability, making it a compelling strategy.

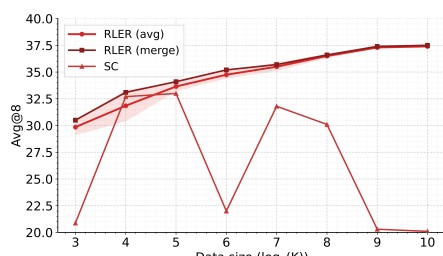

Figure 7: Avg@8 on the test benchmarks under unlabeled data scaling.

## 6 CONCLUSIONS

Reinforcement learning with intrinsic rewards (RLIR), in which the policy model assigns reward signals to its own rollouts, is well suited for sustainable data scaling in unlabeled settings. However, its performance and stability still fall short of RLVR. We attribute this gap to a system bias. By formalizing three metrics: noise rate $\rho_{\text{noise}}$, self-feedback bias $\rho_{\text{selfbias}}$, and symmetry bias $\rho_{\text{symbias}}$, we characterize this bias and identify the key levers for improvement. We therefore propose reinforcement learning with ensembled rewards (RLER). It aggregates diverse policies to build a unified reward-estimation space that jointly optimizes accuracy, unbiasedness, and robustness. Extensive experiments validate RLER's strong performance, along with scalable and stable behavior in practical applications.

## 7 ETHICS STATEMENT

All datasets used in this study are publicly available; no human subjects or annotators were involved. We confirm that our use is consistent with the datasets' licenses and research intent, and that no personally identifiable or harmful content is included. We cite all datasets and related works accordingly.

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

## A    PROOF OF THEOREM IN §3.3

**Setup.**    Given the policy $\pi_\theta$ and the labeling map $\ell : \mathcal{Y} \to \{0, \dots, L-1\}$, define the label probability

$$q_j := \sum_{y_{1:T}:\,\ell(y_{1:T})=j} \prod_{t=1}^{T} \pi_\theta\big(y_t \mid x,\, y_{<t}\big), \qquad q_j \geq 0, \qquad \sum_{j=0}^{L-1} q_j = 1.$$

Let the predicted (MAP) label be $m = \arg\max_j q_j$, and write

$$a := q_t, \qquad b := q_m, \qquad o := 1 - a - b.$$

**Hard vs. Soft rewards.**    For a rollout $y_i$ with label $\ell(y_i)$, define

$$r_i^{\mathrm{H}} = \mathbf{1}\big[\ell(y_i) = m\big], \qquad \mu_{\mathrm{H}} = b, \quad \sigma_{\mathrm{H}}^2 = b(1-b),$$

$$r_i^{\mathrm{S}} = q_{\ell(y_i)}, \qquad S_2 := \sum_j q_j^2, \quad S_3 := \sum_j q_j^3, \qquad \mu_{\mathrm{S}} = S_2, \quad \sigma_{\mathrm{S}}^2 = S_3 - S_2^2.$$

When the intrinsic probabilities are instantiated by the empirical outcome frequencies $q_j = p_j$, the Soft reward $r_i^{\mathrm{S}} = q_{\ell(y_i)}$ reduces to the Frequency-based method, whereas the Hard reward $r_i^{\mathrm{H}} = \mathbf{1}[\ell(y_i) = m]$ coincides with Self-Consistency.

**Aadvantage and correlation criterion.**    For a group $\{r_i\}_{i=1}^{G}$, GRPO uses group-wise standardized advantages

$$\bar{r} = \tfrac{1}{G} \sum_i r_i, \qquad s = \sqrt{\tfrac{1}{G} \sum_i (r_i - \bar{r})^2}, \qquad A_i = \frac{r_i - \bar{r}}{s},$$

Because correlation is affine-invariant, replacing population $(\mu, \sigma)$ by group statistics $(\bar{r}, s)$ leaves the comparison unchanged. Hence, with standardized variables,

$$\mathrm{MSE}(r) = \tfrac{1}{G} \sum_i (A_i - A_i^\star)^2 \;=\; 2\big(1 - \rho(r, r^\star)\big),$$

so that

$$\mathrm{MSE}(r^{\mathrm{S}}) \leq \mathrm{MSE}(r^{\mathrm{H}}) \iff \rho_{\mathrm{S}} \geq \rho_{\mathrm{H}}.$$

When $m \neq t$, both correlations are negative; larger is better.

**Closed forms for $m \neq t$.**    A direct calculation yields

$$\rho_{\mathrm{H}} \;=\; \frac{\mathrm{Cov}(r^{\mathrm{H}}, r^\star)}{\sigma_{\mathrm{H}} \sigma_\star} \;=\; -\sqrt{\frac{ab}{(1-a)(1-b)}}\,,$$

and, using $\mathbb{E}[r^{\mathrm{S}} r^\star] = a^2$,

$$\mathrm{Cov}(r^{\mathrm{S}}, r^\star) = a^2 - a S_2 = -a\,(S_2 - a) < 0, \qquad \rho_{\mathrm{S}} \;=\; -\frac{a\,(S_2 - a)}{\sqrt{a(1-a)\,(S_3 - S_2^2)}}\,.$$

**Tail dispersion monotonicity.**    Fix $(a, b, o)$ induced by $q$. Let $\mathcal{O} = \mathcal{L} \setminus \{m, t\}$ and $s_{\max} = \max_{j \in \mathcal{O}} q_j$. Making the non-majority (tail) mass $o$ more dispersed strictly decreases $S_2 = \sum_j q_j^2$ by convexity of $x^2$ and strictly increases $S_3 - S_2^2$. Therefore $|\rho_{\mathrm{S}}|$ strictly decreases, while $\rho_{\mathrm{H}}$ is unaffected. Hence the *worst case* for $\rho_{\mathrm{S}}$ at fixed $(a, b, o)$ occurs when the tail is fully concentrated, i.e. $s_{\max} = o$.

**Sufficiency.**    In the worst case $s_{\max} = o$,

$$\rho_{\mathrm{S}} - \rho_{\mathrm{H}} \;=\; (a - s_{\max}) \frac{(1-b)\sqrt{a(1-a)}}{\sqrt{ab}\,\sqrt{S_3 - S_2^2}} \;>\; 0 \quad \text{whenever } a \geq s_{\max}.$$

Since tail dispersion only improves $\rho_{\mathrm{S}}$, we have $\rho_{\mathrm{S}} \geq \rho_{\mathrm{H}}$ for all tail configurations whenever $a \geq s_{\max}$.

**Necessity.** If $a < s_{\max}$, concentrate the entire tail mass on a single label so that $s_{\max} = o$. The same expression becomes negative, implying $\rho_S < \rho_H$, i.e. $\mathrm{MSE}(r^S) > \mathrm{MSE}(r^H)$.

**Conclusion.** Under $m \neq t$, the Soft reward is closer to the oracle than the Hard reward *if and only if* $a \geq s_{\max}$, which is the claim of Theorem 1.

## B  MORE EXPERIMENT DETAILS

### B.1  RLER ON ARITHMETIC DATASET

**Experimental Settings** Prior work shows that RL gains are highly sensitive to model pretraining: pretraining on large-scale web corpora can introduce data contamination on popular benchmarks (Wu et al., 2025; Shao et al., 2025). To eliminate contamination effects and cleanly validate our method, we synthesize a decontaminated arithmetic dataset (375k) comprising expressions over operators $\{+, -, //, \%\}$, with $1 - 3$ operators applied to $2 - 6$-digit integers, partitioned into 15 uniformly distributed difficulty groups with increasing hardness.We evaluate on a 500-problem in-distribution, unseen test set. For the model, we use QWEN-2.5-1.5B-INSTRUCT.

We set the number of rollouts to $G=32$, the learning rate to $1 \times 10^{-6}$, the KL regularization coefficient to $\beta=0.001$, the sampling temperature to $0.9$. We train for one epoch. All experiments are conducted on NVIDIA H20 (96 GB).

**Main results** The results of the compared methods on our arithmetic test set are reported in Table 2, and the ablation results for RLER are shown in Table 3. The results show that RLER achieves the best overall performance: Avg@k improves by +14.1 points over the best baseline, and Pass@k suffers the smallest degradation from the pre-RL model. The ablations further validate the necessity of each component.

Table 2: Zero-shot Avg@16 and Pass@16 of QWEN-2.5-1.5B-INSTRUCT on arithmetic test set.

| Method | Avg@16 | Pass@16 |
|---|---|---|
| w/o RL | 41.5 | 89.2 |
| *RLVR* | 93.2 | 95.5 |
| *RLIR* | | |
| LLM-as-a-Judge | 48.3 | 70.6 |
| Self-Consistency | 57.4 | 60.2 |
| Frequency-Based | 56.9 | 62.6 |
| *RLER* ($k = 2$) | 69.2 | 72.2 |
| *RLER* ($k = 4$) | 71.5 | 75.8 |

## C  PROMPT TEMPLATE FOR RLER

```
system_prompt

system_prompt: |
  You are a mathematical reasoning expert. When given a math
    problem, analyze it step by step. First, detail your internal
    reasoning in a <think> block using steps (e.g., "Step 1:",
    "Step 2:", etc.). Then, provide only the final conclusion in an
    <answer> block. Follow this exact format with no extra text:

  <think>
  Step 1: ...
  Step 2: ...
  ...
```

```
</think>
<answer>
...
</answer>
```

Table 3: Ablation results of RLER on arithmetic test set.

| Method | Avg@16 |
|---|---|
| *RLER* | 71.5 |
| *w/o Rollout selection* | |
| Ensemble Interpolation v3 | 69.6 |
| Ensemble Interpolation v2 | 68.2 |
| *w/o Interpolation&Rollout selection* | |
| Ensemble SC | 67.6 |
| Ensemble Freq | 65.8 |
| *w/o Ensemble&Rollout selection* | |
| SC Interpolation v3 | 63.2 |
| SC Interpolation v2 | 61.3 |
| SC Interpolation v1 | 59.8 |
| *w/o all* | |
| SC | 57.4 |
| Freq | 56.9 |

