# OpenReview forum: "Diagnosing and Mitigating Systemic Reward Bias in Self-Rewarding RL"
_ICLR.cc/2026/Conference — Submitted to ICLR 2026_

### Official Review · Reviewer_LJof · 2025-10-26

**Soundness:** 1
**Presentation:** 1
**Contribution:** 2
**Rating:** 2
**Confidence:** 3

**Summary:**

Research has identified a systematic bias in RLIR: models tend to overestimate their own high-confidence outputs, leading to biased and unstable reward estimation, which in turn affects training convergence and performance limits. To mitigate this issue, the authors propose Reinforcement Learning with Ensemble Rewards (RLER), which improves the accuracy, unbiasedness, and robustness of reward estimation by aggregating multiple models and employing an adaptive reward interpolation strategy.

**Strengths:**

1. The analysis of systematic bias is quite comprehensive.
2.  Although ensemble methods are commonly used, the specific implementation and innovative design still contribute meaningfully to the field.

**Weaknesses:**

1. The experiments in this paper are almost entirely based on the Qwen series of models, and the evaluation datasets used (such as MATH500, AMC, AIME, etc.) are highly likely to be contaminated. The authors also acknowledge this issue in the Appendix. Therefore, the experimental results cannot be considered reliable unless validated on more models and datasets with favorable outcomes.
2. Figure 1 needs improvement. The font size in the figure is too small, making it difficult to discern the method the authors intend to illustrate.
3. The training dataset is overly simplistic and lacks diversity.

**Questions:**

1. As mentioned in the Weaknesses section, the authors urgently need to supplement the experiments with more models, training datasets, and evaluation datasets.
2. The computational cost of the ensemble method should be quantified and analyzed.

---

> ### Author Response · Authors · 2025-11-21
> **Response to reviewer LJof (part 1-1)**
>
> Dear Reviewer LJof, we sincerely thank you for your valuable reviews and positive feedback on our submission. Below is our responses to the concerns you raised. We will incorporate these contents into the revised version of our paper, which we believe will help enhance the quality of our submission.
>
> ---
>
> > **“The experiments in this paper are almost entirely based on the Qwen series of models, and the evaluation datasets used (such as MATH500, AMC, AIME, etc.) are highly likely to be contaminated. The authors also acknowledge this issue in the Appendix. Therefore, the experimental results cannot be considered reliable unless validated on more models and datasets with favorable outcomes.”**
>
> > **“The training dataset is overly simplistic and lacks diversity.”**
>
>
> First, we would like to clarify that our use of the Qwen series and math benchmarks closely follows the dominant setups in current RLVR work. Qwen models have been widely validated as strong and effective for reasoning, and mathematical reasoning tasks, due to their importance and verifiability, are the primary focus of most RLVR-based reasoning studies.
>
> We did try several other backbones for RL on math tasks. For example, for Llama-series models on DAPO-MATH-17K, even under a ground-truth-labeled RLVR-style setup, the reward accuracy remained below 10%, making it difficult to obtain meaningful RL improvements. For reasoning-specialized models such as DeepSeek-R1-Distill-series, the extremely long outputs made RL training infeasible under our hardware constraints.
>
> At the same time, we fully understand and share your concern about possible contamination of public training and test sets, especially after reading recent leakage analyses on Qwen models, such as *REASONING OR MEMORIZATION? UNRELIABLE RESULTS OF REINFORCEMENT LEARNING DUE TO DATA CONTAMINATION* [1], which question some recent RL works. To mitigate this, we constructed our own arithmetic dataset and used it to further validate the effectiveness of RLER on less-contaminated data; in particular, under a random-reward setting we observe **no performance gain**, which supports that our improvements do not come from memorization or data leakage.
>
> In response to your comments, we have gone beyond the original Qwen + math setup and **carefully selected additional models, training datasets, and evaluation datasets** to more broadly verify the generality of our method. Below we clarify (i) why we initially focus on verifiable reasoning tasks such as mathematics, (ii) how RLER extends to more general domains where free-text answers can be reliably verified, and (iii) how we verify the generalization of RLER across different models and datasets.
>
> ---
>
> ### (1) Why we start from math-style, verifiable reasoning tasks
>
> * Current RL for LLM reasoning has made the most progress on mathematical and competition-style problems, where answers are *verifiable* and one can obtain **accurate, unbiased rewards**. This is precisely why most RLVR works focus on these tasks.
> * Our main starting point is that **RLVR is bottlenecked by the scarcity of labeled/verifiable data**, which limits continued data scaling in industrial scenarios with abundant unlabeled data and a need for rapid iteration.
> * At the same time, existing *unsupervised* RLIR methods suffer from **systematic bias and instability**, making them hard to deploy in practice.
>
> Therefore, we follow the RLVR setting and choose math-style reasoning benchmarks *deliberately* for two reasons:
>
> 1. **Clean diagnosis of system bias.** With access to accurate oracle rewards, we can precisely quantify noise and bias ($ \rho _ {\text{noise}}, \rho _ {\text{selfbias}}, \rho _ {\text{symbias}} $), and perform controlled ablations to isolate the causal effect of reward/system bias.
> 2. **Faithful evaluation of RLER.** Our adaptive reward design is *specifically* aimed at reducing over-optimization of FP samples (Self-Consistency–like behavior) and recovering low-frequency FN samples (Probability–based behavior), as discussed in Theorem 1. To validate these effects, we need reliable ground-truth rewards.
>
> ---

---

> ### Author Response · Authors · 2025-11-21
> **Response to reviewer LJof (part 1-2)**
>
> ### (2) Cross-task generalization
>
> We fully agree that “math tasks are clear and easy-to-validate” is a strong assumption and is more a property of **RLVR-style tasks** (where answers often come from a finite discrete set) than a fundamental limitation of **RLER**.
>
> Conceptually, RLER only requires that **textual answers can be verified or quantitatively assessed for their semantic equivalence**. In practice, we need a *reasonably reliable verifier* (rule-based or model-based) that can measure **the degree of equivalence between different model-generated textual answers** for the same query.
>
> To support this, we add experiments with **Qwen2.5-7B-Instruct** on **WebInstruct-verified** [2]：a diverse, high-quality dataset to facilitate robust reasoning capabilities across a broad range of domains, extending beyond the commonly studied mathematical problems.
>
> In this setting, an officially provided, well-trained LLM verifier is used to judge output's correctness. We leverage this verifier to assign rewards based on the *equivalence degree between model answers*. We further include a strong RLIR baseline **INTUITOR** [3] and use **MMLU-Pro** [4] (a more robust and challenging massive multi-task understanding benchmark with 12K complex questions across various disciplines) as the evaluation benchmark.
>
> | Method                        | biology | business | chemistry | computer science | engineering | Pass@1 (avg 14 categories) |
> | ----------------------------- | ------- | -------- | --------- | ---------------- | ----------- | -------------------------- |
> | pre-RL                        | 0.5955  | 0.5970   | 0.4823    | 0.5244           | 0.3447      | 0.4872                     |
> | **RLVR**                      | 0.7211  | 0.6296   | 0.5186    | 0.5734           | 0.4239      | 0.5637                     |
> | SC (RLIR)                     | 0.7029  | 0.6071   | 0.5327    | 0.5610           | 0.4056      | 0.5405                     |
> | Judge (RLIR)                  | 0.7071  | 0.6172   | 0.4956    | 0.5756           | 0.3746      | 0.5282                     |
> | **INTUITOR (RLIR)**           | 0.7005  | 0.6123   | 0.5111    | 0.5632           | 0.3987      | 0.5306                     |
> | **RLER (k = 2, $G _ k$ = 8)** | 0.7083  | 0.6159   | 0.5222    | 0.5628           | 0.4130      | 0.5512                     |
>
> We highlight three observations:
>
> 1. **RLER extends beyond math.** RLER *consistently improves* over RLIR baselines, demonstrating that our bias-mitigation ideas generalize to non-mathematical, cross-domain reasoning.
> 2. **RLER remains competitive with RLVR.** RLER significantly closes the gap relative to RLIR baselines, consistent with our math results.
> 3. The use of **WebInstruct-verified** shows that as long as we have a reasonably accurate verifier, it is feasible to quantify the equivalence degree between free-text answers in a wide range of domains and apply RLER on top of such verifiers.
>
> ---

---

> ### Author Response · Authors · 2025-11-21
> **Response to reviewer LJof (part 1-3)**
>
> ### (3) Cross-model and cross-dataset generalization
>
> We also evaluate across **different backbones and training datasets** to test the generality of RLER.
>
> #### (a) Arithmetic dataset with Llama 3.2-3B-Instruct
>
> Using our arithmetic dataset (Appendix B.1) and Llama 3.2-3B-Instruct, we compare pre-RL, RLVR, RLIR baselines (freq, SC, Judge, INTUITOR), and RLER with different ensemble configurations:
>
> | Method                        | Avg@16 | Pass@16 |
> | ----------------------------- | ------ | ------- |
> | pre-RL                        | 30.7   | 80.2    |
> | **RLVR**                      | 70.2   | 87.8    |
> | freq (RLIR)                   | 47.5   | 56.2    |
> | SC (RLIR)                     | 48.4   | 54.4    |
> | Judge (RLIR)                  | 41.7   | 77.3    |
> | **INTUITOR (RLIR)**           | 50.3   | 54.2    |
> | **RLER (k = 2, $G _ k$ = 8)** | 58.7   | 69.8    |
> | **RLER (k = 4, $G _ k$ = 4)** | 62.3   | 74.4    |
> | **RLER (k = 8, $G _ k$ = 2)** | 61.9   | 78.6    |
> | **RLER (k = 4, $G _ k$ = 8)** | 63.9   | 75.2    |
>
> #### (b) BigMath with Llama 3.1-8B-Instruct
>
> We further evaluate on **BigMath** [5] with Llama 3.1-8B-Instruct, again comparing RLVR, RLIR baselines, INTUITOR, and RLER:
>
> | Method                        | Math  | AIME24 | AIME25 | AMC23 | AMC24 | HMMT24 | Avg@8 | Pass@8 |
> | ----------------------------- | ----- | ------ | ------ | ----- | ----- | ------ | ----- | ------ |
> | pre-RL                        | 45.75 | 3.75   | 0      | 19.38 | 12.22 | 0      | 13.52 | 30.20  |
> | **RLVR**                      | 49.95 | 5.00   | 3.33   | 26.25 | 18.13 | 1.25   | 17.32 | 37.31  |
> | SC (RLIR)                     | 42.65 | 2.50   | 2.08   | 24.37 | 12.78 | 0      | 14.06 | 30.85  |
> | Judge (RLIR)                  | 37.22 | 0.42   | 1.25   | 15.62 | 10.83 | 0      | 10.89 | 25.39  |
> | **INTUITOR (RLIR)**           | 46.25 | 3.33   | 2.50   | 21.56 | 12.78 | 0.42   | 14.47 | 31.72  |
> | **RLER (k = 2, $G _ k$ = 8)** | 48.35 | 4.17   | 2.92   | 23.75 | 16.67 | 1.25   | 16.19 | 35.07  |
>
> These results suggest that:
>
> * RLER’s benefits are **not tied to a specific backbone** (we see similar trends on Qwen-Math-7B, Llama-3.2-3B, and Llama-3.1-8B);
> * Nor are they tied to a specific training dataset (DAPO-MATH-17K vs our arithmetic dataset vs BigMath vs WebInstruct-verified).
>
> ---
>
> References:
>
> [1] REASONING OR MEMORIZATION? UNRELIABLE RESULTS OF REINFORCEMENT LEARNING DUE TO DATA CONTAMINATION
>
> [2] WebInstruct-verified: https://huggingface.co/datasets/TIGER-Lab/WebInstruct-verified
>
> [3] *Learning to Reason without External Rewards* (INTUITOR).
>
> [4] MMLU-Pro: https://huggingface.co/datasets/TIGER-Lab/MMLU-Pro
>
> [5] BigMath: https://huggingface.co/datasets/open-r1/Big-Math-RL-Verified-Processed

---

> ### Author Response · Authors · 2025-11-21
> **Response to reviewer LJof (part 2-1)**
>
> > **“The computational cost of the ensemble method should be quantified and analyzed.”**
>
> We very much agree that understanding the trade-off between ensemble size, diversity, noise, and compute is crucial, and we apologize for not making this sufficiently clear in the original submission. In this part, we (A) clarify the *actual* compute cost of RLER vs. single-model RLIR under our main setting, and (B) present a new ensemble-scaling study (Table A/B) that quantifies how performance, diversity, and cost behave as we vary the ensemble size (k).
>
> ---
>
> ### (A) Clarifying the compute cost of RLER vs. RLIR (Table A)
>
> Under our main experimental setting, we fix the total rollout budget per query (Sec. 5.1):
>
> * RLIR (single-model): one policy model generates 16 rollouts.
> * RLER ($k = 2$): two sub-policies each generate 8 rollouts, for the same total ( $G _ {\text{total}} = 16$ ).
>
> Thus, in terms of rollout generation, RLER and RLIR use the *same* total number of rollouts per query.
>
> To analyze the compute–effectiveness trade-off more rigorously, we decompose one RL training step into three parts: 1. rollout generation (forward sampling); 2. reward / advantage computation; 3. policy update (loss + backward pass).
>
> In our implementation, rollout selection is applied *after* advantage computation, i.e., *all* rollouts are generated and scored, but only selected rollouts enter the loss and gradients. This implies: The dominant difference in FLOPs comes from (3) **policy update**.
>
> Let ( $b _ {\text{avg}}$ ) be the average number of selected rollouts per query. Since the backward pass  is roughly a constant factor more expensive than the forward pass (we denote this factor by ( $\gamma \approx 2$ )), we can define a relative compute metric
>
> $$
> C _ {\text{rel}} = \frac{G _ {\text{total}} + \gamma b _ {\text{avg}}}{G _ {\text{base}} + \gamma G _ {\text{base}}}.
> $$
>
>
> We summarize the main-setting results in **Table A**:
>
> **Table A. Relative compute cost under the main setting.**
>
> | Method      | $k$ | $G _ {\text{total}}$ | $b _ {\text{avg}}$ | $\tilde C _ {\text{rel}}$ |
> | ----------- | --: | -------------------: | -----------------: | ------------------------: |
> | RLIR        |   1 |                   16 |               16.0 |                      1.00 |
> | RLER (ours) |   2 |                   16 |               12.0 |                      0.83 |
>
> Under the same rollout budget:
>
> * RLER *update* FLOPs are actually **slightly lower** than the RLIR baseline.
> * The only additional resource cost comes from memory: loading (k) sub-models in parallel increases **VRAM approximately linearly in $k$** (e.g., ≈2× VRAM), which we will explicitly state in the revised paper.
>
> ---

---

> ### Author Response · Authors · 2025-11-21
> **Response to reviewer LJof (part 2-2)**
>
> ### (B) Balancing ensemble size, diversity, and cost: scaling study (Table B)
>
> To directly address your question on how to balance diversity versus noise under a fixed budget, we conduct a **systematic ensemble scaling experiment**, summarized in **Table B**.
>
> We consider two regimes:
>
> 1. **Fixed rollout budget**: total rollouts per query are fixed, and we vary the ensemble size $k$ and per-sub-policy rollouts $G _ k$.
> 2. **Increased budget**: we increase $G _ {\text{total}}$  while keeping the algorithm structure unchanged, to see how RLER scales when more compute is available.
>
> Table B reports, for each configuration: **Avg@8 / Pass@8**, **diversity gain** $\Delta _ {\text{div}}$ (difference between ensemble prediction and single-model average accuracy, see Sec. 5.3), the relative compute proxy $\tilde C _ {\text{rel}}$, and **relative memory** usage.
>
> | Method                                                                      | $k$ | $G _ k$ | Avg@8 | Pass@8 | $\Delta _ {\text{div}}$ |$\tilde C _ {\text{rel}}$ | Memory         |
> | --------------------------------------------------------------------------- | --: | ------: | ----: | -----: | ------------------------: | ------------------------: | -------------- |
> | $ G _ {\text{total}} = 16$     |     |         |       |        |                           |                           |                |
> | RLER                                                               |   2 |       8 |  37.5 |   52.8 |                     0.038 |                        0.83 | $\sim 2\times$ |
> | RLER                                                               |   4 |       4 |  37.6 |   53.5 |                     0.045 |                         0.81 | $\sim 4\times$ |
> | RLER                                                              |   8 |       2 |  37.2 |   54.0 |                     0.051 |                         0.80 | $\sim 8\times$ |
> |$G _ {\text{total}} = 32$ |     |         |       |        |                           |                           |                |
> | RLER                                                                |   4 |       8 |  37.9 |   53.7 |                     0.046 |                         0.73 |$\sim 4\times$ |
>
> ---
>
> #### (B.1) Fixed rollout budget: diversity vs. noise under the same compute
>
> Under the fixed rollout budget $G _ {\text{total}} = 16$, we observe that:
>
> * The **diversity gain** $\Delta _ {\text{div}} $ indeed **increases monotonically** (0.038 → 0.045 → 0.051) as we enlarge $k$, confirming that more sub-models bring more ensemble diversity.
>
> * However, in terms of **final performance**:
>
>   * From $k = 2$ to $k = 4$, Avg@8 only slightly improves (37.5 → 37.6), and Pass@8 improves modestly (52.8 → 53.5);
>   * Increasing to $k = 8$, Pass@8 increases slightly again, but Avg@8 actually **drops** to 37.2, indicating that with very small per-model rollout counts $G _ k = 2$, the variance and noise from each sub-model begin to **offset** the benefits of increased diversity.
>
> In this fixed-budget regime, the three configurations have almost the same FLOPs-level training cost. Therefore, under a **fixed compute budget**, increasing $k$ does *not* significantly change compute cost, but **linearly increases memory usage** (approx. 2×, 4×, 8×), while performance gains **saturate quickly** and even slightly regress beyond.
>
> This suggests a clear **diminishing-return regime**: diversity continues to increase, but the noise introduced by too few rollouts per sub-model erodes the practical benefit.
>
> ---
>
> #### (B.2) Increased budget: scaling in a high-compute regime
>
> To understand the upper bound when more compute is allowed, we include a **high-budget variant** in Table B:
>
> * RLER ($k = 4$, $G _ k = 8$): here, so the forward rollout cost is ≈2× the main setting. This configuration achieves the **best** Avg@8 = 37.9 and Pass@8 = 53.7, with a relatively high diversity gain ( $\Delta _ {\text{div}} = 0.046$ ).
>
> This shows that:
>
> * When the **compute budget is increased**, RLER continues to **scale**: more rollouts and larger ensembles can bring additional gains, consistent with the stable scaling behavior on unlabeled data shown in Figure 7.
> * However, this configuration also has significantly higher memory than the main setting, so we treat it as a high-compute regime rather than our recommended default.
>
> ---

---

> ### Author Response · Authors · 2025-11-21
> **Response to reviewer LJof (part 2-3)**
>
> #### (B.3) Why we choose (k = 2) as the default ensemble size
>
> Our goal with RLER is not only to “slightly improve accuracy on a fixed benchmark”, but to provide a **stable and scalable RLIR alternative** for real-world unlabeled, resource-constrained scenarios. As Figure 7 shows:
>
> > *“In real-world scenarios, the absence of human-annotated labels and limited resources mean we cannot know a priori how much data is needed to reach optimal performance. Compared with RLIR methods, RLER exhibits stably scalable behavior on unlabeled data.”*
>
> In practice, this means we care about:
>
> * **Accuracy and bias reduction** (lower $\rho _ {\text{noise}}$, $\rho _ {\text{selfbias}}$, $\rho _ {\text{symbias}}$);
> * **Stable scaling curves** rather than a single “lucky” point;
> * **Compute and memory feasibility** under realistic constraints.
>
> From this perspective:
>
> * Compared to single-model RLIR, RLER already offers clear improvements in accuracy, system-bias mitigation, and stable scaling behavior.
> * Within RLER, under a fixed rollout budget, $k = 2$ is enough to **substantially suppress system bias** and deliver large gains in both performance and robustness; further increasing $G _ {\text{total}}$ yields very marginal gains while multiplying VRAM usage.
>
> Taking into account **performance gains, stable scaling, FLOPs cost, and memory footprint**, we believe that $k = 2$ is the most reasonable and practically recommended ensemble size.

---

> ### Author Response · Authors · 2025-11-21
> **Response to reviewer LJof (part 3)**
>
> > **“Figure 1 needs improvement. The font size in the figure is too small, making it difficult to discern the method the authors intend to illustrate.”**
>
>  For Figure 1, we have carefully taken your suggestion into account, revised the figure, and updated it in the latest version of the PDF.

---

### Official Review · Reviewer_sZwE · 2025-10-29

**Soundness:** 3
**Presentation:** 3
**Contribution:** 3
**Rating:** 6
**Confidence:** 4

**Summary:**

In this paper, the authors focus on the "systematic bias" problem in RLIR. The challenge of this problem is that models tend to assign a high reward for their own high-confidence (even if wrong) output. The authors systematically analyze the reasons for this problem and propose an ensemble reward strategy. The experiments show that the strategy outperforms baselines and achieves results approaching those of methods that rely on labeled data.

**Strengths:**

1. The motivation of this paper is clear and specific. The authors focus on RLIR's important system bias challenge. This challenge is important for LLM's application.
2. In this paper, the authors provide detailed experiments to validate the effectiveness of the method. They provide sufficient hyperparameters for the experiments. In general, the reproducibility of the method should not be a problem.

**Weaknesses:**

1. The method proposed in this paper is an ensemble method. However, in the experiments, the authors only compare their method with single-model methods. They fail to compare the method with other ensemble methods. This lack makes it hard for the reader to evaluate the effect of the mixed signals. This may raise the question of whether the effectiveness of the method is based on the rationality of the reward or the ensemble strategy itself.

2. The authors only test their method on limited LLM types and sizes. In this paper, the authors test the model only on Qwen, and the scales they choose are 1.5B and 7B. The single model choice and smaller scale may raise the question of whether the method and its strategy are only effective on specific models.

**Questions:**

1. The training phase of the method requires running k models in parallel. This brings k times the computation and memory overhead compared to a single model. The increase in computational cost may limit the application of this method on a large scale or with a larger ensemble size. Authors may want to discuss in detail the trade-off between performance gain and training cost, especially in resource-constrained scenarios.

2. The experiments of this paper mainly focus on mathematical reasoning tasks. The characteristic of these tasks is that the answer is clear and easy-to-validate. However, in broader reasoning tasks, such as code generation and creative writing, the reward signal may be hard to quantify. How the method can be generalized to broader reasoning tasks, or whether the method is focused on solving mathematical reasoning, is an open question.

---

> ### Author Response · Authors · 2025-11-21
> **Response to reviewer sZwE (part 1)**
>
> Dear Reviewer sZwE, we sincerely thank you for your valuable reviews and positive feedback on our submission. Below is our responses to the concerns you raised. We will incorporate these contents into the revised version of our paper, which we believe will help enhance the quality of our submission.
>
> ---
>
> ### 1. On ensemble methods and whether gains come from “mixed signals” or reward design
>
> > **“The method proposed in this paper is an ensemble method. However, in the experiments, the authors only compare their method with single-model methods. They fail to compare the method with other ensemble methods. This lack makes it hard for the reader to evaluate the effect of the mixed signals. This may raise the question of whether the effectiveness of the method is based on the rationality of the reward or the ensemble strategy itself.”**
>
> Thank you very much for this comment. First, we would like to clarify that our use of ensembles is motivated directly by the **systematic analysis of RLIR** in Section 3. We start from the observation that standard RLIR suffers from **system bias** (e.g., single-policy strong policy-reward coupling: over-rewarding the model’s own high-confidence errors), and our goal is to mitigate this bias at the reward level. To this end, we propose to replace single-model self-rewarding with an ensemble, aggregating diverse models to construct a unified, more stable reward space.
>
> Second, in Section 5.3 “VARIANTS & ABLATIONS”, we conduct detailed ablations on the roles and gains of different components in RLER. For example, in Figure 5 (left):
>
> * Starting from a single-model method, adding only the ensemble strategy improves performance by about **10%–25%**;
> * Adding only Soft-Reward Interpolation brings about **+9.1%**;
> * Combining both ensemble and interpolation yields about **+23.1%** overall improvement.
>
> Figure 5 (right) further shows that Soft-Reward Interpolation alone improves the reward accuracy and reduces $ \rho _ {\text{noise}} $ by about **13%**. Together with Figures 4, 6, and 7, these ablations and analyses jointly demonstrate that each component plays a critical role, and that the best performance comes from their combination rather than from “raw ensemble averaging” alone.
>
> In other words, our ablation study itself already compares different combinations of ensemble and reward strategies, and shows that the effectiveness of RLER. We also apologize that we are not aware of prior RLIR-specific “RLER-style” ensemble baselines directly targeting RLIR system bias.
>
> ---

---

> ### Author Response · Authors · 2025-11-21
> **Response to reviewer sZwE (part 2-1)**
>
> ### 2. On generalization across models, scales, and tasks
>
> > **“The authors only test their method on limited LLM types and sizes. In this paper, the authors test the model only on Qwen, and the scales they choose are 1.5B and 7B. The single model choice and smaller scale may raise the question of whether the method and its strategy are only effective on specific models.”**
>
> > **“The experiments of this paper mainly focus on mathematical reasoning tasks. The characteristic of these tasks is that the answer is clear and easy-to-validate. However, in broader reasoning tasks, such as code generation and creative writing, the reward signal may be hard to quantify. How the method can be generalized to broader reasoning tasks, or whether the method is focused on solving mathematical reasoning, is an open question.”**
>
> We sincerely appreciate your insightful comments on the generalization of our method across models and tasks. Below we clarify (i) why we initially focus on verifiable reasoning tasks such as mathematics, (ii) how RLER extends to more general domains where free-text answers can be reliably verified, and (iii) how we verify the generalization of RLER across different models and datasets.
>
> ---
>
> ### (1) Why we start from math-style, verifiable reasoning tasks
>
> * Current RL for LLM reasoning has made the most progress on mathematical and competition-style problems, where answers are *verifiable* and one can obtain **accurate, unbiased rewards**. This is precisely why most RLVR works focus on these tasks.
> * Our main starting point is that **RLVR is bottlenecked by the scarcity of labeled/verifiable data**, which limits continued data scaling in industrial scenarios with abundant unlabeled data and a need for rapid iteration.
> * At the same time, existing *unsupervised* RLIR methods suffer from **systematic bias and instability**, making them hard to deploy in practice.
>
> Therefore, we follow the RLVR setting and choose math-style reasoning benchmarks *deliberately* for two reasons:
>
> 1. **Clean diagnosis of system bias.** With access to accurate oracle rewards, we can precisely quantify noise and bias ($ \rho _ {\text{noise}}, \rho _ {\text{selfbias}}, \rho _ {\text{symbias}} $), and perform controlled ablations to isolate the causal effect of reward/system bias.
> 2. **Faithful evaluation of RLER.** Our adaptive reward design is *specifically* aimed at reducing over-optimization of FP samples (Self-Consistency–like behavior) and recovering low-frequency FN samples (Probability–based behavior), as discussed in Theorem 1. To validate these effects, we need reliable ground-truth rewards.
>
> ---
>
> ### (2) Cross-task generalization
>
> We fully agree that “math tasks are clear and easy-to-validate” is a strong assumption and is more a property of **RLVR-style tasks** (where answers often come from a finite discrete set) than a fundamental limitation of **RLER**.
>
> Conceptually, RLER only requires that **textual answers can be verified or quantitatively assessed for their semantic equivalence**. In practice, we need a *reasonably reliable verifier* (rule-based or model-based) that can measure **the degree of equivalence between different model-generated textual answers** for the same query.

---

> ### Author Response · Authors · 2025-11-21
> **Response to reviewer sZwE (part 2-2)**
>
> To support this, we add experiments with **Qwen2.5-7B-Instruct** on **WebInstruct-verified** [1]：a diverse, high-quality dataset to facilitate robust reasoning capabilities across a broad range of domains, extending beyond the commonly studied mathematical problems.
>
> In this setting, an officially provided, well-trained LLM verifier is used to judge output's correctness. We leverage this verifier to assign rewards based on the *equivalence degree between model answers*. We further include a strong RLIR baseline **INTUITOR** [2] and use **MMLU-Pro** [3] (a more robust and challenging massive multi-task understanding benchmark with 12K complex questions across various disciplines) as the evaluation benchmark.
>
> | Method                        | biology | business | chemistry | computer science | engineering | Pass@1 (avg 14 categories) |
> | ----------------------------- | ------- | -------- | --------- | ---------------- | ----------- | -------------------------- |
> | pre-RL                        | 0.5955  | 0.5970   | 0.4823    | 0.5244           | 0.3447      | 0.4872                     |
> | **RLVR**                      | 0.7211  | 0.6296   | 0.5186    | 0.5734           | 0.4239      | 0.5637                     |
> | SC (RLIR)                     | 0.7029  | 0.6071   | 0.5327    | 0.5610           | 0.4056      | 0.5405                     |
> | Judge (RLIR)                  | 0.7071  | 0.6172   | 0.4956    | 0.5756           | 0.3746      | 0.5282                     |
> | **INTUITOR (RLIR)**           | 0.7005  | 0.6123   | 0.5111    | 0.5632           | 0.3987      | 0.5306                     |
> | **RLER (k = 2, $G _ k$ = 8)** | 0.7083  | 0.6159   | 0.5222    | 0.5628           | 0.4130      | 0.5512                     |
>
> We highlight three observations:
>
> 1. **RLER extends beyond math.** RLER *consistently improves* over RLIR baselines, demonstrating that our bias-mitigation ideas generalize to non-mathematical, cross-domain reasoning.
> 2. **RLER remains competitive with RLVR.** RLER significantly closes the gap relative to RLIR baselines, consistent with our math results.
> 3. The use of **WebInstruct-verified** shows that as long as we have a reasonably accurate verifier, it is feasible to quantify the equivalence degree between free-text answers in a wide range of domains and apply RLER on top of such verifiers.
>
> ---
>
> ### (3) Cross-model and cross-dataset generalization
>
> We also evaluate across **different backbones and training datasets** to test the generality of RLER.
>
> #### (a) Arithmetic dataset with Llama 3.2-3B-Instruct
>
> Using our arithmetic dataset (Appendix B.1) and Llama 3.2-3B-Instruct, we compare pre-RL, RLVR, RLIR baselines (freq, SC, Judge, INTUITOR), and RLER with different ensemble configurations:
>
> | Method                        | Avg@16 | Pass@16 |
> | ----------------------------- | ------ | ------- |
> | pre-RL                        | 30.7   | 80.2    |
> | **RLVR**                      | 70.2   | 87.8    |
> | freq (RLIR)                   | 47.5   | 56.2    |
> | SC (RLIR)                     | 48.4   | 54.4    |
> | Judge (RLIR)                  | 41.7   | 77.3    |
> | **INTUITOR (RLIR)**           | 50.3   | 54.2    |
> | **RLER (k = 2, $G _ k$ = 8)** | 58.7   | 69.8    |
> | **RLER (k = 4, $G _ k$ = 4)** | 62.3   | 74.4    |
> | **RLER (k = 8, $G _ k$ = 2)** | 61.9   | 78.6    |
> | **RLER (k = 4, $G _ k$ = 8)** | 63.9   | 75.2    |
>
> #### (b) BigMath with Llama 3.1-8B-Instruct
>
> We further evaluate on **BigMath** [4] with Llama 3.1-8B-Instruct, again comparing RLVR, RLIR baselines, INTUITOR, and RLER:
>
> | Method                        | Math  | AIME24 | AIME25 | AMC23 | AMC24 | HMMT24 | Avg@8 | Pass@8 |
> | ----------------------------- | ----- | ------ | ------ | ----- | ----- | ------ | ----- | ------ |
> | pre-RL                        | 45.75 | 3.75   | 0      | 19.38 | 12.22 | 0      | 13.52 | 30.20  |
> | **RLVR**                      | 49.95 | 5.00   | 3.33   | 26.25 | 18.13 | 1.25   | 17.32 | 37.31  |
> | SC (RLIR)                     | 42.65 | 2.50   | 2.08   | 24.37 | 12.78 | 0      | 14.06 | 30.85  |
> | Judge (RLIR)                  | 37.22 | 0.42   | 1.25   | 15.62 | 10.83 | 0      | 10.89 | 25.39  |
> | **INTUITOR (RLIR)**           | 46.25 | 3.33   | 2.50   | 21.56 | 12.78 | 0.42   | 14.47 | 31.72  |
> | **RLER (k = 2, $G _ k$ = 8)** | 48.35 | 4.17   | 2.92   | 23.75 | 16.67 | 1.25   | 16.19 | 35.07  |
>
> These results suggest that:
>
> * RLER’s benefits are **not tied to a specific backbone** (we see similar trends on Qwen-Math-7B, Llama-3.2-3B, and Llama-3.1-8B);
> * Nor are they tied to a specific training dataset (DAPO-MATH-17K vs our arithmetic dataset vs BigMath vs WebInstruct-verified).
>
> ---
>
> References:
>
> [1] WebInstruct-verified: https://huggingface.co/datasets/TIGER-Lab/WebInstruct-verified
>
> [2] *Learning to Reason without External Rewards* (INTUITOR).
>
> [3] MMLU-Pro: https://huggingface.co/datasets/TIGER-Lab/MMLU-Pro
>
> [4] BigMath: https://huggingface.co/datasets/open-r1/Big-Math-RL-Verified-Processed

---

> ### Author Response · Authors · 2025-11-21
> **Response to reviewer sZwE (part 3-1)**
>
> > **“The training phase of the method requires running k models in parallel. This brings k times the computation and memory overhead compared to a single model. The increase in computational cost may limit the application of this method on a large scale or with a larger ensemble size. Authors may want to discuss in detail the trade-off between performance gain and training cost, especially in resource-constrained scenarios.”**
>
> We very much agree that understanding the trade-off between ensemble size, diversity, noise, and compute is crucial, and we apologize for not making this sufficiently clear in the original submission. In this part, we (A) clarify the *actual* compute cost of RLER vs. single-model RLIR under our main setting, and (B) present a new ensemble-scaling study (Table A/B) that quantifies how performance, diversity, and cost behave as we vary the ensemble size (k).
>
> ---
>
> ### (A) Clarifying the compute cost of RLER vs. RLIR (Table A)
>
> Under our main experimental setting, we fix the total rollout budget per query (Sec. 5.1):
>
> * RLIR (single-model): one policy model generates 16 rollouts.
> * RLER ($k = 2$): two sub-policies each generate 8 rollouts, for the same total ( $G _ {\text{total}} = 16$ ).
>
> Thus, in terms of rollout generation, RLER and RLIR use the *same* total number of rollouts per query.
>
> To analyze the compute–effectiveness trade-off more rigorously, we decompose one RL training step into three parts: 1. rollout generation (forward sampling); 2. reward / advantage computation; 3. policy update (loss + backward pass).
>
> In our implementation, rollout selection is applied *after* advantage computation, i.e., *all* rollouts are generated and scored, but only selected rollouts enter the loss and gradients. This implies: The dominant difference in FLOPs comes from (3) **policy update**.
>
> Let ( $b _ {\text{avg}}$ ) be the average number of selected rollouts per query. Since the backward pass  is roughly a constant factor more expensive than the forward pass (we denote this factor by ( $\gamma \approx 2$ )), we can define a relative compute metric
>
> $$
> C _ {\text{rel}} = \frac{G _ {\text{total}} + \gamma b _ {\text{avg}}}{G _ {\text{base}} + \gamma G _ {\text{base}}}.
> $$
>
> We summarize the main-setting results in **Table A**:
>
> **Table A. Relative compute cost under the main setting.**
>
> | Method      | $k$ | $G _ {\text{total}}$ | $b _ {\text{avg}}$ | $\tilde C _ {\text{rel}}$ |
> | ----------- | --: | -------------------: | -----------------: | ------------------------: |
> | RLIR        |   1 |                   16 |               16.0 |                      1.00 |
> | RLER (ours) |   2 |                   16 |               12.0 |                      0.83 |
>
> Under the same rollout budget:
>
> * RLER *update* FLOPs are actually **slightly lower** than the RLIR baseline.
> * The only additional resource cost comes from memory: loading (k) sub-models in parallel increases **VRAM approximately linearly in $k$** (e.g., ≈2× VRAM), which we will explicitly state in the revised paper.
>
> ---

---

> ### Author Response · Authors · 2025-11-21
> **Response to reviewer sZwE (part 3-2)**
>
> ### (B) Balancing ensemble size, diversity, and cost: scaling study (Table B)
>
> To directly address your question on how to balance diversity versus noise under a fixed budget, we conduct a **systematic ensemble scaling experiment**, summarized in **Table B**.
>
> We consider two regimes:
>
> 1. **Fixed rollout budget**: total rollouts per query are fixed, and we vary the ensemble size $k$ and per-sub-policy rollouts $G _ k$.
> 2. **Increased budget**: we increase $G _ {\text{total}}$  while keeping the algorithm structure unchanged, to see how RLER scales when more compute is available.
>
> Table B reports, for each configuration: **Avg@8 / Pass@8**, **diversity gain** $\Delta _ {\text{div}}$ (difference between ensemble prediction and single-model average accuracy, see Sec. 5.3), the relative compute proxy $\tilde C _ {\text{rel}}$, and **relative memory** usage.
>
> | Method                                                                      | $k$ | $G _ k$ | Avg@8 | Pass@8 | $\Delta _ {\text{div}}$ |$\tilde C _ {\text{rel}}$ | Memory         |
> | --------------------------------------------------------------------------- | --: | ------: | ----: | -----: | ------------------------: | ------------------------: | -------------- |
> | $ G _ {\text{total}} = 16$     |     |         |       |        |                           |                           |                |
> | RLER                                                               |   2 |       8 |  37.5 |   52.8 |                     0.038 |                        0.83 | $\sim 2\times$ |
> | RLER                                                               |   4 |       4 |  37.6 |   53.5 |                     0.045 |                         0.81 | $\sim 4\times$ |
> | RLER                                                              |   8 |       2 |  37.2 |   54.0 |                     0.051 |                         0.80 | $\sim 8\times$ |
> |$G _ {\text{total}} = 32$ |     |         |       |        |                           |                           |                |
> | RLER                                                                |   4 |       8 |  37.9 |   53.7 |                     0.046 |                         0.73 |$\sim 4\times$ |
>
> ---
>
> #### (B.1) Fixed rollout budget: diversity vs. noise under the same compute
>
> Under the fixed rollout budget $G _ {\text{total}} = 16$, we observe that:
>
> * The **diversity gain** $\Delta _ {\text{div}} $ indeed **increases monotonically** (0.038 → 0.045 → 0.051) as we enlarge $k$, confirming that more sub-models bring more ensemble diversity.
>
> * However, in terms of **final performance**:
>
>   * From $k = 2$ to $k = 4$, Avg@8 only slightly improves (37.5 → 37.6), and Pass@8 improves modestly (52.8 → 53.5);
>   * Increasing to $k = 8$, Pass@8 increases slightly again, but Avg@8 actually **drops** to 37.2, indicating that with very small per-model rollout counts $G _ k = 2$, the variance and noise from each sub-model begin to **offset** the benefits of increased diversity.
>
> In this fixed-budget regime, the three configurations have almost the same FLOPs-level training cost. Therefore, under a **fixed compute budget**, increasing $k$ does *not* significantly change compute cost, but **linearly increases memory usage** (approx. 2×, 4×, 8×), while performance gains **saturate quickly** and even slightly regress beyond.
>
> This suggests a clear **diminishing-return regime**: diversity continues to increase, but the noise introduced by too few rollouts per sub-model erodes the practical benefit.
>
> ---
>
> #### (B.2) Increased budget: scaling in a high-compute regime
>
> To understand the upper bound when more compute is allowed, we include a **high-budget variant** in Table B:
>
> * RLER ($k = 4$, $G _ k = 8$): here, so the forward rollout cost is ≈2× the main setting. This configuration achieves the **best** Avg@8 = 37.9 and Pass@8 = 53.7, with a relatively high diversity gain ( $\Delta _ {\text{div}} = 0.046$ ).
>
> This shows that:
>
> * When the **compute budget is increased**, RLER continues to **scale**: more rollouts and larger ensembles can bring additional gains, consistent with the stable scaling behavior on unlabeled data shown in Figure 7.
> * However, this configuration also has significantly higher memory than the main setting, so we treat it as a high-compute regime rather than our recommended default.
>
> ---

---

> ### Author Response · Authors · 2025-11-21
> **Response to reviewer sZwE (part 3-3)**
>
> #### (B.3) Why we choose (k = 2) as the default ensemble size
>
> Our goal with RLER is not only to “slightly improve accuracy on a fixed benchmark”, but to provide a **stable and scalable RLIR alternative** for real-world unlabeled, resource-constrained scenarios. As Figure 7 shows:
>
> > *“In real-world scenarios, the absence of human-annotated labels and limited resources mean we cannot know a priori how much data is needed to reach optimal performance. Compared with RLIR methods, RLER exhibits stably scalable behavior on unlabeled data.”*
>
> In practice, this means we care about:
>
> * **Accuracy and bias reduction** (lower $\rho _ {\text{noise}}$, $\rho _ {\text{selfbias}}$, $\rho _ {\text{symbias}}$);
> * **Stable scaling curves** rather than a single “lucky” point;
> * **Compute and memory feasibility** under realistic constraints.
>
> From this perspective:
>
> * Compared to single-model RLIR, RLER already offers clear improvements in accuracy, system-bias mitigation, and stable scaling behavior.
> * Within RLER, under a fixed rollout budget, $k = 2$ is enough to **substantially suppress system bias** and deliver large gains in both performance and robustness; further increasing $G _ {\text{total}}$ yields very marginal gains while multiplying VRAM usage.
>
> Taking into account **performance gains, stable scaling, FLOPs cost, and memory footprint**, we believe that $k = 2$ is the most reasonable and practically recommended ensemble size.

---

### Official Review · Reviewer_8jPv · 2025-11-03

**Soundness:** 2
**Presentation:** 2
**Contribution:** 2
**Rating:** 4
**Confidence:** 3

**Summary:**

This paper diagnoses and mitigates system bias in RLIR. The work traces the performance gap between RLIR and methods using RLVR to a system bias where the model incorrectly rewards its own high-confidence outputs. To characterize this, the paper introduces three metrics: reward noise rate, self-feedback bias rate, and symmetry bias rate. Based on this diagnosis, the paper proposes RLER, which aggregates diverse models to create a more accurate and stable reward signal. RLER combines ensemble self-rewarding, adaptive soft-reward interpolation, and a confidence-disagreement balanced rollout selection strategy. Experiments on mathematical reasoning benchmarks show that RLER significantly outperforms RLIR baselines and substantially closes the performance gap with RLVR.

**Strengths:**

- This work introduces three distinct metrics, which provide a clear and quantitative framework for analyzing system bias.
- The decoupling experiment is a well-designed study that effectively isolates the impact of each metric on training dynamics.
- The paper presents extensive experiments with strong results that convincingly support its claims.

**Weaknesses:**

Lack of precision in key definitions and insufficient details regarding experimental setups, particularly in Section 3. For example, the "attained reward as r_i" (line 150) is unclear whether it is a model prediction or a noisy label. "hard-reward" and "soft-reward" are mentioned without explanation in the main text. The motivation for choosing the three specific metrics, which are central to the paper's contribution, could also be explained in detail. "Findings 2" concludes that over-reward is more detrimental than under-reward, and a "Further analysis" is mentioned to support this. However, the text does not elaborate on this analysis, and the connection between the cited figures (Fig. 2(b) and Fig. 3(e)) and the claim about a "near-orthogonal gradient bias" is not explained The setup for experiments in Figure 3 is not described with sufficient detail.  the paper states that an ensemble of k=2 models is used but fails to describe how these models are chosen or initialized. A small ensemble size (k=2) is also not fully justified. p_selfbias^true(x) and p_selfbias^err(x), without defining what "true" and "err" mean in this context.

**Questions:**

See weakness and:

The main experiments use an ensemble of 2 models. What is the rationale for this specific choice? How does the performance, computational overhead, and model diversity of RLER scale on the main benchmarks as ensemble number is increased beyond 2?

---

> ### Author Response · Authors · 2025-11-21
> **Response to reviewer 8jPv (part 1-1)**
>
> Dear Reviewer 8jPv, we sincerely thank you for your valuable reviews and positive feedback on our submission. Below is our responses to the concerns you raised.
>
> **1. On the clarity and precision of definitions and experimental details**
>
> We are very grateful that you took the time to carefully read our paper and patiently point out places where our explanations are ambiguous or imprecise. We would like to briefly clarify these issues here, and we will carefully revise the corresponding parts in the next version of the paper according to your suggestions.
>
> ---
>
> #### (1-1) On the definition of “attained reward as $r _ i$”
>
> > **“For example, the ‘attained reward as $r _ i$ (line 150) is unclear whether it is a model prediction or a noisy label.”**
>
> Thank you very much for pointing out this ambiguity. In Section 3, the intended meaning of the “attained reward” $r _ i$ is: the actual reward signal received by the RLIR self-reward module on the $i$-th rollout.
> In the revised version, we will explicitly describe $r _ i$ as “the reward label under the current policy model’s prediction” and clearly define it at its first occurrence as a potentially noisy label that may differ from the oracle reward $r _ i ^ {*} $ because of system bias.
>
> ---
>
> #### (1-2) On the meaning of “hard-reward” and “soft-reward”
>
> > **“‘hard-reward’ and ‘soft-reward’ are mentioned without explanation in the main text.”**
>
> We apologize for not providing clear definitions in the main text. In Section 2 (RELATED WORKS), we conceptually categorize existing RLIR methods by (i) the information source of self-reward estimation and (ii) how sharply they refine the answer distribution into a reward distribution (binary 0/1 vs. continuous), leading to three families: (i) Self-Consistency, (ii) Probability–based, and (iii) LLM-as-a-Judge.
>
> In our notation:
>
> * **Hard reward** $r _ i ^ {\text{H}}$ is a **discrete, thresholded reward**: we first map the final answer of a rollout to a verifiable outcome (e.g., correct vs. incorrect), then assign a fixed scalar reward, typically in ${0, 1}$.
>
>   * On math benchmarks, this is based on symbolic equivalence of the solution: in RLVR it uses the ground-truth label; in RLIR it is often based on the majority vote (Self-Consistency) or a judge model’s verdict.
> * **Soft reward** $r _ i ^ {\text{S}}$ is a **continuous reward**. In probability-based methods, it is typically computed as the *expected correctness* of a response according to the policy’s confidence/entropy.
>
> Our view is that these two types of rewards are **internally aligned in their objective** (both aim to maximize agreement between the answer distribution and the reward distribution), but they suffer from complementary failure modes:
>
> * hard rewards tend to be *overly sharp*, so high-confidence FP samples are strongly over-rewarded and their errors accumulate;
> * soft rewards tend to suffer from many low-frequency FN cases, where correct but low-confidence answers are under-rewarded.
>
> In RLER, we use the ensemble-calibrated confidence in the answer distribution to construct an interpolated reward $r _ i ^ {\alpha}$ that combines the strengths of both.
>
> Theoretically, **Theorem 1** analyzes, under different conditions, the regions where $r _ i ^ {\text{S}}$ vs. $r _ i ^ {\text{H}}$ are closer to the oracle reward $r _ i ^ {*} $. Based on this, our Adaptive Soft-Reward Interpolation adopts the following form:
>
> $$
> r _ i ^ {\alpha} = (1 - \alpha) r _ i ^ {\text{H}} + \alpha r _ i ^ {\text{S}}, \quad \alpha \in [0, 1].
> $$
>
> In the revised manuscript, we will add precise definitions of hard/soft rewards in Section 3, together with a short intuitive explanation, and explicitly connect them to Theorem 1.
>
> ---

---

> ### Author Response · Authors · 2025-11-21
> **Response to reviewer 8jPv (part 1-2)**
>
> #### (1-3) On the motivation for the three metrics and their roles
>
> > **“The motivation for choosing the three specific metrics, which are central to the paper’s contribution, could also be explained in detail.”**
>
> We agree that the motivation behind our metric design needs to be stated more explicitly. Our starting point is to conceptually characterize RLIR’s *system bias* into three complementary failure modes:
>
> 1. **Overall reward noise magnitude**: how often the self-reward deviates from the oracle reward (observed behavior);
> 2. **Policy–reward coupling strength**: the degree of policy-reward coupling (root cause);
> 3. **Over- vs. under-reward asymmetry**: whether errors are dominated by FP (incorrect answers being over-rewarded) or FN (correct answers being under-rewarded) (noise structure).
>
> Accordingly, we design:
>
> * $ \rho _ {\text{noise}} $: the overall inconsistency rate between self-reward and oracle reward, capturing the *magnitude* of reward noise;
> * $ \rho _ {\text{selfbias}} $: the degree of **policy–reward coupling**, i.e., whether the reward tends to favor the model’s own predictions;
> * $ \rho _ {\text{symbias}} $: by decomposing errors into FP/FN, this measures the **asymmetry** between over-reward and under-reward.
>
> We choose these three metrics because:
>
> * Together, they form a **compact but complementary set**: in the decoupling experiments (Figure 3), we show that each metric has a causal impact on RL dynamics when varied in isolation. Increasing $ \rho_{\text{noise}} $ slows convergence and lowers the final performance, while $ \rho_{\text{selfbias}} $ mainly shifts the asymptotic performance under a fixed noise level. Under the same $ \rho_{\text{noise}} $, policy-independent random noise is about **30% less harmful** than policy-dependent self-rewarding noise. Finally, $ \rho_{\text{symbias}} $ captures the strong asymmetry of reward noise: FP hurt much more than FN — correcting FN rewards yields roughly **20% better** final performance than correcting FP rewards by the same amount.
>
> * They are also **directly actionable**: different components of RLER target different failure modes:
>   * the interpolation mainly reduces $ \rho _ {\text{noise}} $ and $ \rho _ {\text{symbias}} $ by smoothing hard errors and recovering under-rewarded cases;
>   * balanced allocation&selection strategies are designed to alleviate $ \rho _ {\text{selfbias}} $;
>   * the ensemble mechanism, as the most critical factor, aggregates diverse models into a **unified reward space** and helps improve all three metrics simultaneously.
>
> In the revised version, we will add a short subsection in Section 3 explicitly explaining this design motivation.
>
> ---
>
> #### (1-4) On “Findings 2”, over-reward vs. under-reward, and near-orthogonal gradient bias
>
> > **“‘Findings 2’ concludes that over-reward is more detrimental than under-reward, and a ‘Further analysis’ is mentioned to support this. However, the text does not elaborate on this analysis, and the connection between the cited figures (Fig. 2(b) and Fig. 3(e)) and the claim about a ‘near-orthogonal gradient bias’ is not explained.”**
>
>  As mentioned above, one of our key motivations for introducing $ \rho _ {\text{symbias}} $ is exactly to capture the asymmetry between over-reward and under-reward. “Findings 2” empirically shows that, under comparable noise levels, over-reward errors (FPs) are more harmful than under-reward errors (FNs).
>
> Intuitively:
>
> * Over-rewarding incorrect answers induces gradient updates that move the policy **towards a wrong mode** in the answer space;
> * Under-rewarding correct answers mainly reduces the *strength* of positive updates but does not directly push the policy towards incorrect modes.
>
> In Figures 3(e), we decompose the gradients induced by FP (over-reward) and FN (under-reward) errors relative to the oracle gradient. We find that FN-induced gradients are largely parallel to the oracle direction and mainly reduce the magnitude along the correct direction. In contrast, FP-induced gradients add components that are nearly orthogonal to the oracle gradient, effectively rotating the update direction away from the optimum. Under iterative updates, such orthogonal bias accumulates and leads to much larger deviation than parallel bias, which explains why over-reward (FP) is more detrimental than under-reward (FN).
>
> We acknowledge that this reasoning is not sufficiently spelled out in the current text. In the revision, we will expand the “Further analysis” paragraph under “Findings 2” with a concise explanation of this near-orthogonal gradient bias;
>
>
> ---

---

> ### Author Response · Authors · 2025-11-21
> **Response to reviewer 8jPv (part 1-3)**
>
> #### (1-5) On the setup of Figure 3 and how the $k = 2$ ensemble is constructed
>
> > **“The setup for experiments in Figure 3 is not described with sufficient detail. The paper states that an ensemble of k = 2 models is used but fails to describe how these models are chosen or initialized.”**
>
> Thank you for pointing out this missing detail. In our implementation, we adopt a population-based strategy:
>
> * We start from the **same backbone** to initialize the sub-models in the ensemble;
> * During training, our rollout allocation & selection strategy distributes different subsets of updates to different sub-models, which gradually induces diversity in the population and mitigates the strong system bias that a single model would have;
> * For deployment, we apply model merging technique (ENSEMBLE-TO-SINGLE) to merge the sub-models back into a single model for deployment.
>
> Figures 4–7 empirically show that this strategy yields a final model with better accuracy, unbiasedness, and robustness, and, importantly, stably scalable behavior on unlabeled data, which is particularly crucial compared to RLIR in real-world settings.
>
> We will add a detailed description of this population-based training and the $k = 2$ ensemble construction in the experimental setup for Figure 3 in the revised version.
>
> ---
>
> #### (1-6) On the definitions of $p _ {\text{selfbias}} ^ {\text{true}}(x)$ and $p _ {\text{selfbias}} ^ {\text{err}}(x)$
>
> > **“$p _ {\text{selfbias}} ^ {\text{true}}(x)$ and $p _ {\text{selfbias}} ^ {\text{err}}(x)$, without defining what ‘true’ and ‘err’ mean in this context.”**
>
> Thank you for pointing out this unclear notation. In our definitions, for a given input $x$, we consider the model’s predicted label
> ( $m = \arg\max _ j p _ j$ ), and then split cases by whether this prediction is correct:
>
> * $ \rho _ {\text{selfbias}} ^ {\text{true}}(x)$ denotes the policy–reward coupling strength when the model prediction is correct. Ideally, this should be high: the reward system should strongly support correct predictions;
> * $ \rho _ {\text{selfbias}} ^ {\text{err}}(x)$ denotes the policy–reward coupling strength when the model prediction is incorrect. Ideally, this should be low: the reward system should not strongly reinforce wrong predictions.
>
> We further decompose $ \rho _ {\text{selfbias}} $ into these two refined quantities mainly based on Findings 3 & 4, which show that high self-bias on incorrect predictions is a key source of instability and performance bottlenecks in RLIR. In our experiments, we use $ \rho _ {\text{selfbias}} ^ {\text{true}}$ and $ \rho _ {\text{selfbias}} ^ {\text{err}}$ to verify that RLER reduces harmful self-bias on errors while improving useful self-bias on correct predictions.
>
> In the revised manuscript, we will explicitly define “true” and “err” at the first introduction of these symbols and add a short explanation of why this decomposition is useful for analyzing and mitigating RLIR system bias.

---

> ### Author Response · Authors · 2025-11-21
> **Response to reviewer 8jPv (part 2-1)**
>
> **2. On the choice of ensemble size: trade-off between cost and effectiveness**
>
> > **“A small ensemble size (k=2) is also not fully justified.”**
>
> > **“The main experiments use an ensemble of 2 models. What is the rationale for this specific choice? How does the performance, computational overhead, and model diversity of RLER scale on the main benchmarks as ensemble number is increased beyond 2?”**
>
> We very much agree that understanding the trade-off between ensemble size, diversity, noise, and compute is crucial, and we apologize for not making this sufficiently clear in the original submission. In this part, we (A) clarify the *actual* compute cost of RLER vs. single-model RLIR under our main setting, and (B) present a new ensemble-scaling study (Table A/B) that quantifies how performance, diversity, and cost behave as we vary the ensemble size (k).
>
> ---
>
> ### (A) Clarifying the compute cost of RLER vs. RLIR (Table A)
>
> Under our main experimental setting, we fix the total rollout budget per query (Sec. 5.1):
>
> * RLIR (single-model): one policy model generates 16 rollouts.
> * RLER ($k = 2$): two sub-policies each generate 8 rollouts, for the same total ( $G _ {\text{total}} = 16$ ).
>
> Thus, in terms of rollout generation, RLER and RLIR use the *same* total number of rollouts per query.
>
> To analyze the compute–effectiveness trade-off more rigorously, we decompose one RL training step into three parts: 1. rollout generation (forward sampling); 2. reward / advantage computation; 3. policy update (loss + backward pass).
>
> In our implementation, rollout selection is applied *after* advantage computation, i.e., *all* rollouts are generated and scored, but only selected rollouts enter the loss and gradients. This implies: The dominant difference in FLOPs comes from (3) **policy update**.
>
> Let ( $b _ {\text{avg}}$ ) be the average number of selected rollouts per query. Since the backward pass  is roughly a constant factor more expensive than the forward pass (we denote this factor by ( $\gamma \approx 2$ )), we can define a relative compute metric
>
> $$
> C _ {\text{rel}} = \frac{G _ {\text{total}} + \gamma b _ {\text{avg}}}{G _ {\text{base}} + \gamma G _ {\text{base}}}.
> $$
>
>
> We summarize the main-setting results in **Table A**:
>
> **Table A. Relative compute cost under the main setting.**
>
> | Method      | $k$ | $G _ {\text{total}}$ | $b _ {\text{avg}}$ | $\tilde C _ {\text{rel}}$ |
> | ----------- | --: | -------------------: | -----------------: | ------------------------: |
> | RLIR        |   1 |                   16 |               16.0 |                      1.00 |
> | RLER (ours) |   2 |                   16 |               12.0 |                      0.83 |
>
> Under the same rollout budget:
>
> * RLER *update* FLOPs are actually **slightly lower** than the RLIR baseline.
> * The only additional resource cost comes from memory: loading (k) sub-models in parallel increases **VRAM approximately linearly in $k$** (e.g., ≈2× VRAM), which we will explicitly state in the revised paper.
>
> ---
>
> ### (B) Balancing ensemble size, diversity, and cost: scaling study (Table B)
>
> To directly address your question on how to balance diversity versus noise under a fixed budget, we conduct a **systematic ensemble scaling experiment**, summarized in **Table B**.
>
> We consider two regimes:
>
> 1. **Fixed rollout budget**: total rollouts per query are fixed, and we vary the ensemble size $k$ and per-sub-policy rollouts $G _ k$.
> 2. **Increased budget**: we increase $G _ {\text{total}}$  while keeping the algorithm structure unchanged, to see how RLER scales when more compute is available.

---

> ### Author Response · Authors · 2025-11-21
> **Response to reviewer 8jPv (part 2-2)**
>
> Table B reports, for each configuration: **Avg@8 / Pass@8**, **diversity gain** $\Delta _ {\text{div}}$ (difference between ensemble prediction and single-model average accuracy, see Sec. 5.3), the relative compute proxy $\tilde C _ {\text{rel}}$, and **relative memory** usage.
>
> | Method                                                                      | $k$ | $G _ k$ | Avg@8 | Pass@8 | $\Delta _ {\text{div}}$ |$\tilde C _ {\text{rel}}$ | Memory         |
> | --------------------------------------------------------------------------- | --: | ------: | ----: | -----: | ------------------------: | ------------------------: | -------------- |
> | $ G _ {\text{total}} = 16$     |     |         |       |        |                           |                           |                |
> | RLER                                                               |   2 |       8 |  37.5 |   52.8 |                     0.038 |                        0.83 | $\sim 2\times$ |
> | RLER                                                               |   4 |       4 |  37.6 |   53.5 |                     0.045 |                         0.81 | $\sim 4\times$ |
> | RLER                                                              |   8 |       2 |  37.2 |   54.0 |                     0.051 |                         0.80 | $\sim 8\times$ |
> |$G _ {\text{total}} = 32$ |     |         |       |        |                           |                           |                |
> | RLER                                                                |   4 |       8 |  37.9 |   53.7 |                     0.046 |                         0.73 |$\sim 4\times$ |
>
> ---
>
> #### (B.1) Fixed rollout budget: diversity vs. noise under the same compute
>
> Under the fixed rollout budget $G _ {\text{total}} = 16$, we observe that:
>
> * The **diversity gain** $\Delta _ {\text{div}} $ indeed **increases monotonically** (0.038 → 0.045 → 0.051) as we enlarge $k$, confirming that more sub-models bring more ensemble diversity.
>
> * However, in terms of **final performance**:
>
>   * From $k = 2$ to $k = 4$, Avg@8 only slightly improves (37.5 → 37.6), and Pass@8 improves modestly (52.8 → 53.5);
>   * Increasing to $k = 8$, Pass@8 increases slightly again, but Avg@8 actually **drops** to 37.2, indicating that with very small per-model rollout counts $G _ k = 2$, the variance and noise from each sub-model begin to **offset** the benefits of increased diversity.
>
> In this fixed-budget regime, the three configurations have almost the same FLOPs-level training cost. Therefore, under a **fixed compute budget**, increasing $k$ does *not* significantly change compute cost, but **linearly increases memory usage** (approx. 2×, 4×, 8×), while performance gains **saturate quickly** and even slightly regress beyond.
>
> This suggests a clear **diminishing-return regime**: diversity continues to increase, but the noise introduced by too few rollouts per sub-model erodes the practical benefit.
>
> ---
>
> #### (B.2) Increased budget: scaling in a high-compute regime
>
> To understand the upper bound when more compute is allowed, we include a **high-budget variant** in Table B:
>
> * RLER ($k = 4$, $G _ k = 8$): here, so the forward rollout cost is ≈2× the main setting. This configuration achieves the **best** Avg@8 = 37.9 and Pass@8 = 53.7, with a relatively high diversity gain ( $\Delta _ {\text{div}} = 0.046$ ).
>
> This shows that:
>
> * When the **compute budget is increased**, RLER continues to **scale**: more rollouts and larger ensembles can bring additional gains, consistent with the stable scaling behavior on unlabeled data shown in Figure 7.
> * However, this configuration also has significantly higher memory than the main setting, so we treat it as a high-compute regime rather than our recommended default.
>
> ---

---

> ### Author Response · Authors · 2025-11-21
> **Response to reviewer 8jPv (part 2-3)**
>
> #### (B.3) Why we choose (k = 2) as the default ensemble size
>
> Our goal with RLER is not only to “slightly improve accuracy on a fixed benchmark”, but to provide a **stable and scalable RLIR alternative** for real-world unlabeled, resource-constrained scenarios. As Figure 7 shows:
>
> > *“In real-world scenarios, the absence of human-annotated labels and limited resources mean we cannot know a priori how much data is needed to reach optimal performance. Compared with RLIR methods, RLER exhibits stably scalable behavior on unlabeled data.”*
>
> In practice, this means we care about:
>
> * **Accuracy and bias reduction** (lower $\rho _ {\text{noise}}$, $\rho _ {\text{selfbias}}$, $\rho _ {\text{symbias}}$);
> * **Stable scaling curves** rather than a single “lucky” point;
> * **Compute and memory feasibility** under realistic constraints.
>
> From this perspective:
>
> * Compared to single-model RLIR, RLER already offers clear improvements in accuracy, system-bias mitigation, and stable scaling behavior.
> * Within RLER, under a fixed rollout budget, $k = 2$ is enough to **substantially suppress system bias** and deliver large gains in both performance and robustness; further increasing $G _ {\text{total}}$ yields very marginal gains while multiplying VRAM usage.
>
> Taking into account **performance gains, stable scaling, FLOPs cost, and memory footprint**, we believe that $k = 2$ is the most reasonable and practically recommended ensemble size.

---

### Official Review · Reviewer_fLzJ · 2025-11-06

**Soundness:** 2
**Presentation:** 3
**Contribution:** 3
**Rating:** 6
**Confidence:** 3

**Summary:**

The paper analyzes the system bias issues in RLIR (Reinforcement Learning with Intrinsic Rewards) through reward bias magnitude, policy-reward coupling strength, and imbalance magnitude. To address the systemic bias issues, the paper proposes RLER (Reinforcement Learning with Ensembled Rewards). Specifically, RLER replaces the single-model self-rewarding with an ensemble, aggregating diverse models to construct a unified stable reward space that guides the ensemble to improve collaboratively. The paper uses Qwen2.5-Math-7B as the backbone model and trains it with DAPO-MATH-17K dataset. Experiments demonstrate that RLER outperforms the RLIR baseline across multiple metrics and achieves results very close to RLVR (Reinforcement Learning with Verifiable Rewards).

**Strengths:**

1. The paper provides a novel approach to analyze the system bias of RLIR. The paper characterizes RLIR's noise, characterizing RLIR’s noise, coupling, and over/under-reward asymmetry with three metrics and validates their causal roles through experiments.

2. The experiment section of this paper is comprehensive. It includes not only the overall improvement rate but also the analytical indicators mentioned earlier and a full range of ablation experiments.

**Weaknesses:**

1. The reward design in this paper is relatively complex, increasing parameter tuning costs while potentially causing gradient instability. Additionally, the reward/scoring module exhibits high sensitivity to batch composition, sampling temperature, and normalization strategies, where even minor missteps can amplify training fluctuations.

2. This paper assumes that free-text answers can be mapped to discrete categories, upon which rewards are calculated. While this approach may be feasible for tasks like mathematics, it struggles to adapt to open-ended tasks. Furthermore, the experimental data presented in the paper are exclusively from mathematics datasets, failing to demonstrate advantages across other task domains.

3. RLER relies on the diversity of candidate results derived from model diversity. With too few models, the advantages of ensemble learning are not fully realized, while too many models introduce noise. The quantitative analysis in this paper is insufficient in this regard.

4. The paper only includes LLM-as-a-Judge and typical RLIR as the baseline model. It does not incorporate recently designed improvements specifically addressing RLIR limitations as experimental baselines.

**Questions:**

Model diversity brings ensemble benefits but may also amplify noise. How to optimally balance diversity versus noise within a fixed budget?

---

> ### Author Response · Authors · 2025-11-21
> **Response to reviewer fLzJ (part 1-1)**
>
> Dear Reviewer fLzJ, we sincerely thank you for your valuable reviews and positive feedback on our submission. Below is our responses to the concerns you raised. We will incorporate these contents into the revised version of our paper, which we believe will help enhance the quality of our submission.
>
> > **“The reward design in this paper is relatively complex, increasing parameter tuning costs while potentially causing gradient instability. Additionally, the reward/scoring module exhibits high sensitivity to batch composition, sampling temperature, and normalization strategies, where even minor missteps can amplify training fluctuations.”**
>
> We sincerely appreciate your insightful comments on the stability and tuning aspects of RLIR, and we apologize for not explaining these points clearly enough in the original submission. For part 1, below we clarify (i) what needs to be tuned in RLER, and (ii) how we empirically evaluate and improve stability w.r.t. rollout selection, sampling temperature, and the interpolation bounds.
>
> ---
>
> ### (1) On parameter tuning cost: only a small adaptive set
>
> Although the reward design of RLER may appear complex, the number of *method-specific* hyperparameters that actually require tuning is small:
>
> * the **ensemble size** $k$ (discussed in detail in part 3 of our response with Table B);
> * the **batch-level interpolation bounds** $L _ {\min}^{(k)}$, $L _ {\max}^{(k)}$ used in Adaptive Soft-Reward Interpolation.
>
> All other components (e.g., confidence normalization, selection weights) are *fully data-driven* and computed adaptively from the current batch statistics.
>
> ---
>
> ### (2) On training stability: RLER is designed to mitigate RLIR’s inherent instability
>
> We fully agree with your observation that RLIR is intrinsically prone to instability: it is essentially RL under system noise generated by the policy itself. As Figure 7 shows, standard RLIR methods often exhibit unstable scaling behavior as unlabeled data increases.
>
> RLER is *explicitly* designed to address this:
>
> * **Ensemble self-rewarding** decouples reward estimation from a single model’s system bias and reduces $ \rho _ {\text{noise}}, \rho _ {\text{selfbias}}, \rho _ {\text{symbias}} $.
> * **Adaptive Soft-Reward Interpolation** further stabilizes the reward signal by combining hard and soft rewards based on unified, ensemble-level confidence, as validated in Figures 3–6.
>
> Below we address the specific sensitivity you mentioned.
>
> ---
>
> ### (3) Sensitivity to batch composition and rollout selection
>
> You pointed out that the reward/scoring module may be sensitive to batch composition. To address this, we explicitly analyzed our **Confidence–Disagreement Balanced Rollout Selection** in Figure 6.
>
> As shown in Figure 6 (right), based on controlled statistics:
>
> * when the ensemble majority prediction is correct ($m _ {\text{EC}} = t$), our method selects a much larger fraction of rollouts (≈75%);
> * when $m _ {\text{EC}} \neq t$, RLER **effectively discards false-positive (FP) rollouts**, reducing the reward noise $ \rho _ {\text{noise}} $ by about **15%** compared to “select all”.
>
> This means that, instead of amplifying the instability due to batch variation, RLER uses ensemble disagreement *and* confidence to actively filter out high-confidence FP samples and retain scarce FN samples, thereby **stabilizing gradient updates** under noisy RLIR conditions.
>
> ---

---

> ### Author Response · Authors · 2025-11-21
> **esponse to reviewer fLzJ (part 1-2)**
>
> ### (4) Robustness to sampling temperature
>
> Motivated by your comment on sampling temperature, we conducted additional experiments on **Llama 3.2 3B Instruct** (arithmetic dataset) and **Qwen2.5-Math-7B** (DAPO -17K), comparing Self-Consistency (SC) and RLER under $t \in {0.5, 0.7, 0.9 \text{ (default)}, 1.0}$.
> For each setting, we report: **Pass@1**, **checkpoint accuracy variance** (acc variance across ±5 checkpoints around the best one, reflecting training stability), and **entropy**.
>
> The detailed numbers are shown in the table below (each cell: *Pass@1 / Var / Ent*):
>
> | Model                 | Method | t = 0.5             | t = 0.7             | t = 0.9 (default)   | t = 1.0             |
> | --------------------- | ------ | ------------------- | ------------------- | ------------------- | ------------------- |
> | Llama 3.2 3B Instruct | SC     | 51.9 / 78.92 / 1.09 | 51.3 / 33.90 / 1.83 | 48.4 / 3.44 / 1.90  | 49.3 / 6.77 / 3.66  |
> |                       | RLER   | 57.2 / 4.26 / 0.70  | 58.3 / 2.78 / 0.92  | 58.7 / 2.50 / 1.39  | 59.6 / 3.70 / 2.37  |
> | Qwen2.5-Math-7B       | SC     | 32.4 / 72.50 / 0.08 | 31.8 / 38.00 / 0.10 | 33.0 / 32.11 / 0.17 | 32.7 / 34.90 / 0.32 |
> |                       | RLER   | 37.6 / 2.10 / 0.05  | 36.9 / 1.23 / 0.09  | 37.5 / 2.10 / 0.06  | 38.8 / 1.23 / 0.09  |
>
> From the table above, we summarize the key trends:
>
> * **RLER is robust across temperatures.** For both models, RLER’s Pass@1 remains consistently high and improves over SC for all temperatures.
> * **RLER shows much lower checkpoint variance.** Compared to SC, RLER’s checkpoint acc variance is significantly reduced, indicating substantially more *stable* training, consistent with the scaling behavior in Figure 7.
> * **RLER has lower entropy.** SC, due to strong single-model system bias, tends to produce high-entropy FP samples, which leads to wrong-direction updates and instability (aligning with Findings 2 and Figures 2(b), 3(e)); RLER, by contrast, yields lower-entropy, more calibrated predictions.
>
> These results empirically support that **RLER *reduces*, rather than increases, sensitivity to the sampling temperature and training noise**.
>
> ---
>
> ### (5) Robustness to $L _ {\min}^{(k)}$, $L _ {\max}^{(k)}$ in Adaptive Soft-Reward Interpolation
>
> In RLER, Adaptive Soft-Reward Interpolation uses batch-level confidence ranges to modulate the hard/soft mixture:
>
> * For each source model, we normalize confidence within $[L _ {\min}^{(k)}, L _ {\max}^{(k)}]$.
> * Intuitively, items above $L _ {\max}^{(k)}$ are “sufficiently confident” and can rely more on hard rewards, while items below $L _ {\min}^{(k)}$ receive stronger soft-reward smoothing (Theorem 1 in Sec. 3.3 discusses when soft rewards are closer to the oracle than hard rewards).
>
> We added a new sensitivity study varying $L _ {\min}^{(k)} \in {0.10, 0.20, 0.30}$ and $L _ {\max}^{(k)} \in {0.50, 0.60, 0.70}$. The resulting Avg@8 on the six math benchmarks is summarized below:
>
> | $L _ {\min}^{(k)}$ | $L _ {\max}^{(k)} = 0.50$ | $L _ {\max}^{(k)} = 0.60$ (default) | $L _ {\max}^{(k)} = 0.70$ |
> | ------------------ | ------------------------- | ----------------------------------- | ------------------------- |
> | 0.10               | 37.0                      | 37.3                                | 37.1                      |
> | 0.20 (default)     | 37.4                      | **37.5**                            | 37.3                      |
> | 0.30               | 36.9                      | 37.2                                | 37.0                      |
>
> We observe that:
>
> * performance varies only within a **narrow band** (around $\pm 0.2\sim0.3$ points around the default);
> * the default setting $L _ {\min}^{(k)} = 0.20$, $L _ {\max}^{(k)} = 0.60$ is slightly but consistently near the optimum.
>
> This shows that RLER is **not highly sensitive** to the exact choice of these bounds. In practice, we will recommend the ranges
> $L _ {\min}^{(k)} \in [0.10, 0.40]$, $L _ {\max}^{(k)} \in [0.50, 0.80]$, which all yield very similar performance in our experiments.
>
> ---
>
> ### (6) Why the interpolation itself improves stability
>
> Although the interpolation may look complex, it is explicitly designed to *improve* robustness. First, the **ensemble** creates a unified and more stable reward space (Figure 4), substantially reducing $ \rho_{\text{noise}} $, especially on erroneous high-confidence rollouts. Within this space, Adaptive Soft-Reward Interpolation leverages *batch-level* answer confidence and *rollout-level* flatness to adjust the hard/soft weighting, **dampening high-confidence errors** while preserving true positives. Empirically, interpolation alone brings ≈13% improvement in reward accuracy (Figure 5), and RLER reduces $ \rho_{\text{noise}} $ by about **50%** compared to RLIR baselines (Figure 4). Overall, RLER does **not** require heavy manual tuning and **substantially improves stability** over standard RLIR across both training steps and sampling temperatures.

---

> ### Author Response · Authors · 2025-11-21
> **Response to reviewer fLzJ (part 2-1)**
>
> > **“This paper assumes that free-text answers can be mapped to discrete categories, upon which rewards are calculated. While this approach may be feasible for tasks like mathematics, it struggles to adapt to open-ended tasks. Furthermore, the experimental data presented in the paper are exclusively from mathematics datasets, failing to demonstrate advantages across other task domains.”**
>
> > **“The paper only includes LLM-as-a-Judge and typical RLIR as the baseline model. It does not incorporate recently designed improvements specifically addressing RLIR limitations as experimental baselines.”**
>
> We sincerely appreciate your insightful comments on the *applicability* of our method beyond math and the *completeness* of our baselines. Below we clarify (i) why we initially focus on verifiable reasoning tasks such as mathematics, (ii) how RLER extends to more general domains where free-text answers can be reliably verified, and (iii) how we verify the generalization of RLER across different models and datasets, and add strong RLIR baselines such as INTUITOR.
>
> ---
>
> ### (1) Why we start from math-style, verifiable reasoning tasks
>
> We fully agree that assuming free-text answers can be mapped to discrete correctness categories is restrictive for truly open-ended generation. However, this assumption is **not specific to RLER** but rather intrinsic to the **RLVR-style reasoning tasks** that our work is built upon:
>
> * Current RL for LLM reasoning has made the most progress on **mathematical and competition-style problems**, where answers are *verifiable* and one can obtain **accurate, unbiased rewards**. This is precisely why most RLVR works focus on these tasks.
> * Our main starting point is that **RLVR is bottlenecked by the scarcity of labeled/verifiable data**, which limits continued data scaling in industrial scenarios with abundant unlabeled data and a need for rapid iteration.
> * At the same time, existing *unsupervised* RLIR methods suffer from **systematic bias and instability**, making them hard to deploy in practice.
>
> Therefore, we follow the RLVR setting and choose math-style reasoning benchmarks *deliberately* for two reasons:
>
> 1. **Clean diagnosis of system bias.** With access to accurate oracle rewards, we can precisely quantify noise and bias ($ \rho _ {\text{noise}}, \rho _ {\text{selfbias}}, \rho _ {\text{symbias}} $), and perform controlled ablations to isolate the causal effect of reward/system bias.
> 2. **Faithful evaluation of RLER.** Our  adaptive reward design are *specifically* aimed at reducing over-optimization of FP samples (SC) and reducing low-frequency FN samples (probability-based), as discussed in Theorem 1. To validate this, we need reliable ground-truth rewards.
>
> ---
>
> ### (2) On the “mapping free-text answers to discrete categories” assumption and applicability to broader tasks
>
> We fully agree that *“mapping free-text answers to discrete categories”* is a strong assumption. In fact, this is a property of **RLVR-style tasks** (where answers often come from a finite discrete set, e.g., numeric solutions in math), rather than a fundamental limitation of **RLER**.
>
> Conceptually, RLER only requires that textual answers can be *verified or quantitatively assessed for their semantic equivalence*. We just need a *reasonably reliable verifier* (rule-based or model-based) that can measure **the degree of equivalence between different model-generated textual answers** for the same query.

---

> ### Author Response · Authors · 2025-11-21
> **Response to reviewer fLzJ (part 2-2)**
>
> To support this, we add experiments with **Qwen2.5-7B-Instruct** on **WebInstruct-verified** [1]：a diverse, high-quality dataset to facilitate robust reasoning capabilities across a broad range of domains, extending beyond the commonly studied mathematical problems.
>
> In this setting, an officially provided, well-trained LLM verifier is used to judge output's correctness. We leverage this verifier to assign rewards based on the *equivalence degree between model answers*. We further include a strong RLIR baseline **INTUITOR** [2] and use **MMLU-Pro** [3] (a more robust and challenging massive multi-task understanding benchmark with 12K complex questions across various disciplines) as the evaluation benchmark.
>
> | Method                        | biology | business | chemistry | computer science | engineering | Pass@1 (avg 14 categories) |
> | ----------------------------- | ------- | -------- | --------- | ---------------- | ----------- | -------------------------- |
> | pre-RL                        | 0.5955  | 0.5970   | 0.4823    | 0.5244           | 0.3447      | 0.4872                     |
> | **RLVR**                      | 0.7211  | 0.6296   | 0.5186    | 0.5734           | 0.4239      | 0.5637                     |
> | SC (RLIR)                     | 0.7029  | 0.6071   | 0.5327    | 0.5610           | 0.4056      | 0.5405                     |
> | Judge (RLIR)                  | 0.7071  | 0.6172   | 0.4956    | 0.5756           | 0.3746      | 0.5282                     |
> | **INTUITOR (RLIR)**           | 0.7005  | 0.6123   | 0.5111    | 0.5632           | 0.3987      | 0.5306                     |
> | **RLER (k = 2, $G _ k$ = 8)** | 0.7083  | 0.6159   | 0.5222    | 0.5628           | 0.4130      | 0.5512                     |
>
> We highlight three observations:
>
> 1. **RLER extends beyond math**: On MMLU-Pro, RLER *consistently improves* over RLIR baselines, demonstrating that our bias-mitigation ideas generalize to **non-mathematical, cross-domain reasoning**.
> 2. **RLER remains competitive with RLVR**: RLER significantly closes the gap relative to RLIR baselines, consistent with our math results.
> 3. The use of **WebInstruct-verified** shows that as long as we have a reasonably accurate verifier, *mapping free-text answers to discrete equivalence categories is feasible* in a broad range of domains.
>
> ---
>
> ### (3) Cross-model&dataset generalization
>
> We evaluate across **different backbones and training datasets** to test the generality of RLER.
>
> #### (a) Arithmetic dataset with Llama 3.2-3B-Instruct
>
> Using our arithmetic dataset (Appendix B.1) and Llama 3.2-3B-Instruct, we compare pre-RL, RLVR, RLIR baselines (freq, SC, Judge, INTUITOR), and RLER with different ensemble configurations:
>
> | Method                        | Avg@16 | Pass@16 |
> | ----------------------------- | ------ | ------- |
> | pre-RL                        | 30.7   | 80.2    |
> | **RLVR**                      | 70.2   | 87.8    |
> | freq (RLIR)                   | 47.5   | 56.2    |
> | SC (RLIR)                     | 48.4   | 54.4    |
> | Judge (RLIR)                  | 41.7   | 77.3    |
> | **INTUITOR (RLIR)**           | 50.3   | 54.2    |
> | **RLER (k = 2, $G _ k$ = 8)** | 58.7   | 69.8    |
> | **RLER (k = 4, $G _ k$ = 4)** | 62.3   | 74.4    |
> | **RLER (k = 8, $G _ k$ = 2)** | 61.9   | 78.6    |
> | **RLER (k = 4, $G _ k$ = 8)** | 63.9   | 75.2    |
>
>
> #### (b) BigMath with Llama 3.1-8B-Instruct
>
> We further evaluate on **BigMath** [4] with Llama 3.1-8B-Instruct, again comparing RLVR, RLIR baselines and RLER:
>
> | Method                        | Math  | AIME24 | AIME25 | AMC23 | AMC24 | HMMT24 | Avg@8 | Pass@8 |
> | ----------------------------- | ----- | ------ | ------ | ----- | ----- | ------ | ----- | ------ |
> | pre-RL                        | 45.75 | 3.75   | 0      | 19.38 | 12.22 | 0      | 13.52 | 30.20  |
> | **RLVR**                      | 49.95 | 5.00   | 3.33   | 26.25 | 18.13 | 1.25   | 17.32 | 37.31  |
> | SC (RLIR)                     | 42.65 | 2.50   | 2.08   | 24.37 | 12.78 | 0      | 14.06 | 30.85  |
> | Judge (RLIR)                  | 37.22 | 0.42   | 1.25   | 15.62 | 10.83 | 0      | 10.89 | 25.39  |
> | **INTUITOR (RLIR)**           | 46.25 | 3.33   | 2.50   | 21.56 | 12.78 | 0.42   | 14.47 | 31.72  |
> | **RLER (k = 2, $G _ k$ = 8)** | 48.35 | 4.17   | 2.92   | 23.75 | 16.67 | 1.25   | 16.19 | 35.07  |
>
> This suggests that:
>
> * RLER’s benefits are **not tied to a specific backbone** (we see similar trends on Qwen-Math-7B, Llama-3.2-3B, and Llama-3.1-8B);
> * Nor are they tied to a specific training dataset (DAPO-MATH-17K vs our arithmetic dataset vs BigMath vs WebInstruct).
>
>
> ---
>
> References:
>
> [1] WebInstruct-verified: https://huggingface.co/datasets/TIGER-Lab/WebInstruct-verified
>
> [2] *Learning to Reason without External Rewards* (INTUITOR).
>
> [3] MMLU-Pro: https://huggingface.co/datasets/TIGER-Lab/MMLU-Pro
>
> [4] BigMath: https://huggingface.co/datasets/open-r1/Big-Math-RL-Verified-Processed

---

> ### Author Response · Authors · 2025-11-21
> **Response to reviewer fLzJ (part 3-1)**
>
> > **“RLER relies on the diversity of candidate results derived from model diversity. With too few models, the advantages of ensemble learning are not fully realized, while too many models introduce noise. The quantitative analysis in this paper is insufficient in this regard.”**
>
> > **“Model diversity brings ensemble benefits but may also amplify noise. How to optimally balance diversity versus noise within a fixed budget?”**
>
> We very much agree that understanding the trade-off between ensemble size, diversity, noise, and compute is crucial, and we apologize for not making this sufficiently clear in the original submission. In this part, we (A) clarify the *actual* compute cost of RLER vs. single-model RLIR under our main setting, and (B) present a new ensemble-scaling study (Table A/B) that quantifies how performance, diversity, and cost behave as we vary the ensemble size (k).
>
> ---
>
> ### (A) Clarifying the compute cost of RLER vs. RLIR (Table A)
>
> Under our main experimental setting, we fix the total rollout budget per query (Sec. 5.1):
>
> * RLIR (single-model): one policy model generates 16 rollouts.
> * RLER ($k = 2$): two sub-policies each generate 8 rollouts, for the same total ( $G _ {\text{total}} = 16$ ).
>
> Thus, in terms of rollout generation, RLER and RLIR use the *same* total number of rollouts per query.
>
> To analyze the compute–effectiveness trade-off more rigorously, we decompose one RL training step into three parts: 1. rollout generation (forward sampling); 2. reward / advantage computation; 3. policy update (loss + backward pass).
>
> In our implementation, rollout selection is applied *after* advantage computation, i.e., *all* rollouts are generated and scored, but only selected rollouts enter the loss and gradients. This implies: The dominant difference in FLOPs comes from (3) **policy update**.
>
> Let ( $b _ {\text{avg}}$ ) be the average number of selected rollouts per query. Since the backward pass  is roughly a constant factor more expensive than the forward pass (we denote this factor by ( $\gamma \approx 2$ )), we can define a relative compute metric
>
> $$
> C _ {\text{rel}} = \frac{G _ {\text{total}} + \gamma b _ {\text{avg}}}{G _ {\text{base}} + \gamma G _ {\text{base}}}.
> $$
>
>
> We summarize the main-setting results in **Table A**:
>
> **Table A. Relative compute cost under the main setting.**
>
> | Method      | $k$ | $G _ {\text{total}}$ | $b _ {\text{avg}}$ | $\tilde C _ {\text{rel}}$ |
> | ----------- | --: | -------------------: | -----------------: | ------------------------: |
> | RLIR        |   1 |                   16 |               16.0 |                      1.00 |
> | RLER (ours) |   2 |                   16 |               12.0 |                      0.83 |
>
> Under the same rollout budget:
>
> * RLER *update* FLOPs are actually **slightly lower** than the RLIR baseline.
> * The only additional resource cost comes from memory: loading (k) sub-models in parallel increases **VRAM approximately linearly in $k$** (e.g., ≈2× VRAM), which we will explicitly state in the revised paper.
>
> ---
>
> ### (B) Balancing ensemble size, diversity, and cost: scaling study (Table B)
>
> To directly address your question on how to balance diversity versus noise under a fixed budget, we conduct a **systematic ensemble scaling experiment**, summarized in **Table B**.
>
> We consider two regimes:
>
> 1. **Fixed rollout budget**: total rollouts per query are fixed, and we vary the ensemble size $k$ and per-sub-policy rollouts $G _ k$.
> 2. **Increased budget**: we increase $G _ {\text{total}}$  while keeping the algorithm structure unchanged, to see how RLER scales when more compute is available.

---

> ### Author Response · Authors · 2025-11-21
> **Response to reviewer fLzJ (part 3-2)**
>
> Table B reports, for each configuration: **Avg@8 / Pass@8**, **diversity gain** $\Delta _ {\text{div}}$ (difference between ensemble prediction and single-model average accuracy, see Sec. 5.3), the relative compute proxy $\tilde C _ {\text{rel}}$, and **relative memory** usage.
>
> | Method                                                                      | $k$ | $G _ k$ | Avg@8 | Pass@8 | $\Delta _ {\text{div}}$ |$\tilde C _ {\text{rel}}$ | Memory         |
> | --------------------------------------------------------------------------- | --: | ------: | ----: | -----: | ------------------------: | ------------------------: | -------------- |
> | $ G _ {\text{total}} = 16$     |     |         |       |        |                           |                           |                |
> | RLER                                                               |   2 |       8 |  37.5 |   52.8 |                     0.038 |                        0.83 | $\sim 2\times$ |
> | RLER                                                               |   4 |       4 |  37.6 |   53.5 |                     0.045 |                         0.81 | $\sim 4\times$ |
> | RLER                                                              |   8 |       2 |  37.2 |   54.0 |                     0.051 |                         0.80 | $\sim 8\times$ |
> |$G _ {\text{total}} = 32$ |     |         |       |        |                           |                           |                |
> | RLER                                                                |   4 |       8 |  37.9 |   53.7 |                     0.046 |                         0.73 |$\sim 4\times$ |
>
> ---
>
> ### (B.1) Fixed rollout budget: diversity vs. noise under the same compute
>
> Under the fixed rollout budget $G _ {\text{total}} = 16$, we observe that:
>
> * The **diversity gain** $\Delta _ {\text{div}} $ indeed **increases monotonically** (0.038 → 0.045 → 0.051) as we enlarge $k$, confirming that more sub-models bring more ensemble diversity.
>
> * However, in terms of **final performance**:
>
>   * From $k = 2$ to $k = 4$, Avg@8 only slightly improves (37.5 → 37.6), and Pass@8 improves modestly (52.8 → 53.5);
>   * Increasing to $k = 8$, Pass@8 increases slightly again, but Avg@8 actually **drops** to 37.2, indicating that with very small per-model rollout counts $G _ k = 2$, the variance and noise from each sub-model begin to **offset** the benefits of increased diversity.
>
> In this fixed-budget regime, the three configurations have almost the same FLOPs-level training cost. Therefore, under a **fixed compute budget**, increasing $k$ does *not* significantly change compute cost, but **linearly increases memory usage** (approx. 2×, 4×, 8×), while performance gains **saturate quickly** and even slightly regress beyond.
>
> This suggests a clear **diminishing-return regime**: diversity continues to increase, but the noise introduced by too few rollouts per sub-model erodes the practical benefit.
>
> ---
>
> ### (B.2) Increased budget: scaling in a high-compute regime
>
> To understand the upper bound when more compute is allowed, we include a **high-budget variant** in Table B:
>
> * RLER ($k = 4$, $G _ k = 8$): here, so the forward rollout cost is ≈2× the main setting. This configuration achieves the **best** Avg@8 = 37.9 and Pass@8 = 53.7, with a relatively high diversity gain ( $\Delta _ {\text{div}} = 0.046$ ).
>
> This shows that:
>
> * When the **compute budget is increased**, RLER continues to **scale**: more rollouts and larger ensembles can bring additional gains, consistent with the stable scaling behavior on unlabeled data shown in Figure 7.
> * However, this configuration also has significantly higher memory than the main setting, so we treat it as a high-compute regime rather than our recommended default.
>
> ---

---

> ### Author Response · Authors · 2025-11-21
> **Response to reviewer fLzJ (part 3-3)**
>
> ### (B.3) Why we choose (k = 2) as the default ensemble size
>
> Our goal with RLER is not only to “slightly improve accuracy on a fixed benchmark”, but to provide a **stable and scalable RLIR alternative** for real-world unlabeled, resource-constrained scenarios. As Figure 7 shows:
>
> > *“In real-world scenarios, the absence of human-annotated labels and limited resources mean we cannot know a priori how much data is needed to reach optimal performance. Compared with RLIR methods, RLER exhibits stably scalable behavior on unlabeled data.”*
>
> In practice, this means we care about:
>
> * **Accuracy and bias reduction** (lower $\rho _ {\text{noise}}$, $\rho _ {\text{selfbias}}$, $\rho _ {\text{symbias}}$);
> * **Stable scaling curves** rather than a single “lucky” point;
> * **Compute and memory feasibility** under realistic constraints.
>
> From this perspective:
>
> * Compared to single-model RLIR, RLER already offers clear improvements in accuracy, system-bias mitigation, and stable scaling behavior.
> * Within RLER, under a fixed rollout budget, $k = 2$ is enough to **substantially suppress system bias** and deliver large gains in both performance and robustness; further increasing $G _ {\text{total}}$ yields very marginal gains while multiplying VRAM usage.
>
> Taking into account **performance gains, stable scaling, FLOPs cost, and memory footprint**, we believe that $k = 2$ is the most reasonable and practically recommended ensemble size.

---

> ### Author Response · Authors · 2025-11-26
>
> Thank you very much for taking the time to read our rebuttal; we really appreciate your careful evaluation and constructive comments throughout the process. If there are any remaining concerns or points that you feel are still insufficiently addressed, we would be more than happy to provide additional clarification.
>
> If you find that the additional analyses and experiments in the rebuttal address your earlier concerns and further strengthen the contribution of this work, we would be sincerely grateful if you could consider a stronger recommendation.
>
> Thank you again for your time and support.

---

### Meta-Review · Area_Chair_5rLA · 2026-01-06

**Summary:**

The reviewers focus on the computational cost and memory cost associated with the model ensembles. Meanwhile, how the method scales as more models are ensembled is also a key concern. Some reviewers also raise concerns about the diversity of tasks and datasets. Details on experiments should also be enhanced further to make the choice more plausible.

**Reviewer Concerns:**

The computation costs related to hyperparameter tuning and model update are clarified, despite the fact that the memory cost still exists with the increase in the number of ensembled models. The scaling of this method will still be outstanding because k=2 can hardly be considered persuasive that the method scales well.

**Reviewer Scores:**

Reviewer fLzJ and Reviewer sZwE provide positive feedback on this work in the original, and they will maintain the score as long as some concerns are addressed. Reviewer 8jPv will increase his score because some details are provided in the rebuttal. Reviewer LJof will also increase his score due to some concerns being addressed, such as backbones, typos, and computation cost, but some key points will make the reviewer still hold the negative opinion, for example, the training dataset and why k=2 as the ensemble size in this task.

---

### Decision · Program_Chairs · 2026-01-26

Reject